# ADJUSTING PREDICTION MODEL THROUGH WASSERSTEIN GEODESIC FOR CAUSAL INFERENCE

**Yuguang Yan**[1], **Haolin Yang**[1], **Zecong Chen**[1], **Weilin Chen**[1], **Ruichu Cai**[1,2]*, **Zhifeng Hao**[3]

[1] School of Computer Science, Guangdong University of Technology, Guangzhou, China,
[2] Pazhou Laboratory (Huangpu), Guangzhou, China,
[3] College of Science, Shantou University, Shantou, China.

## ABSTRACT

Causal inference estimates the treatment effect by comparing the potential outcomes of the treated and control groups. Due to the existence of confounders, the distributions of treated and control groups are imbalanced, resulting in limited generalization ability of the outcome prediction model, *i.e.*, the prediction model trained on one group cannot perform well on the other group. To tackle this, existing methods usually adjust confounders to learn balanced representations for aligning the distributions. However, these methods could suffer from the over-balancing issue that predictive information about outcomes is removed during adjustment. In this paper, we propose to adjust the outcome prediction model to improve its generalization ability on both groups simultaneously, so that the over-balancing issue caused by confounder adjustment can be avoided. To address the challenge of large distribution discrepancy between groups during model adjustment, we propose to generate intermediate groups through the Wasserstein geodesic, which smoothly connects the control and treated groups. Based on this, we gradually adjust the outcome prediction model between consecutive groups by a self-training paradigm. To further enhance the performance of the model, we filter the generated samples to select high-quality samples for learning. We provide the theoretical analysis regarding our method, and demonstrate the effectiveness of our method on several benchmark datasets in terms of multiple evaluation metrics.

## 1 INTRODUCTION

Causal inference from observational data has been widely used in many real-world applications to evaluate the effects of a treatment (Yazdani & Boerwinkle, 2014; Varian, 2016), such as healthcare (Foster et al., 2011) and recommendation (Sato et al., 2020; Luo et al., 2024; Gao et al., 2024; Wang et al., 2024b). Based on the Rubin-Neyman potential outcome framework (Rubin, 1974; Splawa-Neyman et al., 1990), the treatment effect can be estimated by comparing the potential outcomes under treatment and no treatment, where the outcome models are independently trained on the treated and control groups to estimate the respective potential outcomes (Künzel et al., 2019). Nevertheless, due to confounders that affect both treatment assignment and potential outcomes, the covariates of the control and treated groups follow significantly different distributions, thereby limiting the generalization ability of the outcome prediction models. For instance, patients receiving surgery (*i.e.*, the treated group) typically have more severe conditions than patients not receiving surgery (*i.e.*, the control group). As a result, the prediction model trained on the treated group fails to accurately predict outcomes for the control group.

To tackle the above confounding bias, existing methods usually adjust and align confounders by learning latent representations, so that the distributions of the control and treated groups are balanced, and the outcome prediction models can be generalized between groups (Robins et al., 2000; Shalit et al., 2017; Johansson et al., 2018). However, these methods could excessively balance the distributions of two groups while ignoring discriminative information which are vital for outcome prediction, suffering from the over-balancing issue (Yao et al., 2018; Du et al., 2021). In the extreme

---

*Corresponding author: cairuichu@gmail.com

situation where two distributions collapse to a single point, the distributions are perfectly balanced, while the predictive information for outcomes is completely eliminated. Although a compromise between distribution balancing and outcome prediction can be considered (Shalit et al., 2017), it is heuristic and remains under-explored how to achieve a promising trade-off.

In this paper, we seek to adjust outcome prediction models rather than covariates, so that the over-balancing issue can be avoided. It is non-trivial to adjust the outcome prediction model from one group to the other group due to the significant distribution shift between them. To address this, we propose a method named G-learner, which generates a sequence of intermediate groups that drift gradually from the control group to the treated group, and then gradually adjusts the outcome prediction model trained on one group to the other. Specifically, we establish an optimal transport model between the control and treated groups, which induces a geodesic path between these two distributions in the sense of the Wasserstein metric (Villani, 2021). Such a Wasserstein geodesic is derived from the theory of optimal transport (Monge, 1781; Kantorovitch, 1958), which has demonstrated a powerful ability to characterize data distribution (Courty et al., 2016; 2017). Based on this, we exploit data distributions of the control and treated groups to generate intermediate groups, which smoothly connect the two groups and are beneficial for model adjustment.

Through the generated intermediate group sequence derived from the Wasserstein geodesic, we employ a self-training strategy to transfer the prediction model trained on the control group towards the treated group and vice versa. Since the model adjustment is performed between two consecutive distributions, it is feasible to effectively transfer the prediction model trained on the previous distribution to the next distribution. To further improve the generalization ability of our model on different groups, we design a data filtering method to select generated samples according to the confidence of the predicted results, so that reliable generated samples are leveraged to transfer the prediction models.

We summarize our major contributions as follows.

- We propose an intermediate group generation method based on the Wasserstein geodesic to smoothly connect the control and treated groups, so that one group can be gradually shifted to the other group.

- We gradually adjust the outcome prediction model between the control and treated groups through the generated intermediate groups, which improves the generalization ability of the outcome prediction model.

- We theoretically analyze the effect estimation error of our proposed method, and empirically demonstrate the effectiveness of our method on several benchmark datasets.

## 2 RELATED WORKS

Over the past decades, a variety of methods have been proposed to address the issue of confounding bias. Reweighting aims to reweight samples to reduce the distribution shift between the groups (Hainmueller, 2012). For example, Rosenbaum & Rubin (1983) adopt the inverse of propensity scores as the sample weights, and Kuang et al. (2017) learn sample weights by aligning the moments between the control and treated groups. Representation learning methods adjust covariates by learning balanced representations, so that the distributions of the control and treated groups in the embedding space can be aligned (Johansson et al., 2016). Shalit et al. (2017) trains a neural network to learn balanced representations, where the distribution shift is measured by the Integral Probability Metric (IPM). In (Kazemi & Ester, 2024), the Kullback-Leibler divergence is adopted to measure the distribution shift. However, these methods could excessively balance the covariates while ignoring the discriminative information for potential outcome prediction, which is known as the issue of over-balancing and hampers the performance of causal inference (Johansson et al., 2018; Zhao et al., 2019). Different from them that adjust the covariates by learning sample weights or representations, we adjust the outcome prediction model to improve the generalization ability on different groups, avoiding the over-balancing issue in existing methods.

Recently, optimal transport has also been employed for causal inference (Yan et al., 2024; Wang et al., 2026). Optimal transport studies how to transport masses from one distribution to another distribution with a minimized cost (Monge, 1781; Kantorovitch, 1958; Villani, 2021). Existing studies

have demonstrated the powerful ability of optimal transport to model data distribution (Courty et al., 2016; 2017; Adler & Lunz, 2018). Inspired by this, optimal transport is applied in causal inference to reduce the confounding bias (Li et al., 2021). In (Yan et al., 2024), sample weights are learned based on a semi-relaxed optimal transport model between control and treated groups. In (Shalit et al., 2017; Wang et al., 2023), IPM is implemented by the Wasserstein distance to measure the distribution shift between groups, and balanced representations are learned to minimize the Wasserstein distance. Wang et al. (2025b) utilizes Wasserstein distance to minimize the distribution gap between biased observations and the target population for effective debiasing. These methods are still under the paradigm of reweighting or representation learning, in which the optimal transport cost is minimized to align the distributions of the control and treated groups. Different from them, we generate intermediate groups between distributions through the Wasserstein geodesic, which smoothly connects the control and treated groups in the sense of the metric defined by optimal transport. Based on the generated intermediate groups, we can gradually adjust the outcome prediction model from one group to another.

Self-training methods (Grandvalet & Bengio, 2004; Zou et al., 2019; Gao et al., 2021) have been applied to the problem of semi-supervised learning and domain adaptation (Nigam et al., 2000; Grandvalet & Bengio, 2004; Han et al., 2019). Amini & Gallinari (2002) presents a semi-supervised algorithm that improves classification by iteratively training on unlabeled data with pseudo labels. Han et al. (2019) calibrates predictive uncertainties between source and target domains using Bayesian neural networks, enabling reliable pseudo-labeling and effective unsupervised domain adaptation. He et al. (2024) proposes to generate intermediate domains and applies gradual self-training, significantly improving domain adaptation when intermediate domains are scarce. Compared with them, we leverage self-training to learn counterfactual outcome prediction models, and design a data filtering method to select reliable samples for training. Moreover, we provide a theoretical analysis regarding the effect estimation error of our self-training methods.

## 3 PROBLEM STATEMENT

We adopt the Neyman–Rubin potential outcomes framework (Rubin, 1974; Splawa-Neyman et al., 1990). Let $\{(\mathbf{x}_i, y_i, t_i)\}_{i=1}^n$ denote $n$ samples drawn from the joint distribution of covariates $X$, treated assignment $T$, and outcome $Y$, where $\mathbf{x}_i \in \mathcal{X}$ denotes the covariates of the $i$-th sample, $t_i \in \mathcal{T} = \{0, 1\}$ indicates the treatment assignment, and $y_i \in \mathbb{R}$ denotes the observed factual outcome under the treatment $t_i$. The observed outcome $Y$ is the potential outcome $Y(t)$ corresponding to the actually received treatment $t$. The control group ($t = 0$) received no treatment is denoted as $\{\mathbf{x}_{0,i}\}_{i=1}^{n_0}$, and the treated group ($t = 1$) received the treatment is denoted as $\{\mathbf{x}_{1,j}\}_{j=1}^{n_1}$, where $n_0$ and $n_1$ are the numbers of the samples in two groups, respectively. The first subscript $0/1$ is the group index, and the second subscript $i/j$ is the sample index.

For given $\mathbf{x} \in \mathcal{X}$, our objective is to estimate the conditional average treatment effect (CATE) as follows:

$$\tau(\mathbf{x}) = \mathbb{E}[Y(1) - Y(0)|X = \mathbf{x}]. \tag{1}$$

To facilitate the identification and estimation of causal effects within the framework outlined above, we typically invoke the following key assumptions:

**Assumption 1** (Stable Unit Treatment Value Assumption). The potential outcomes for any sample do not vary with the treatments assigned to other samples, and for each sample, there are no different forms or versions of each treatment value which leads to different potential outcomes.

**Assumption 2** (Ignorability). Conditional on covariates, the treatment assignment is independent of potential outcomes: $T \perp\!\!\!\perp Y(t)|X$.

**Assumption 3** (Positivity). Conditional on covariates, the treatment assignment is not deterministic: $0 < p(T = t|X = \mathbf{x}) < 1$.

Under these assumptions, $\tau(\mathbf{x})$ can be represented by $f(\mathbf{x}, t)$ as follow:

$$\tau(\mathbf{x}) = \mathbb{E}[Y(1) - Y(0)|X = \mathbf{x}] = f(\mathbf{x}, 1) - f(\mathbf{x}, 0) \tag{2}$$

which quantifies how the effects vary with the treatment received by an individual. To predict potential outcomes, we define a function $h(\mathbf{x}, t)$ to estimate the ground-true ITE as follow:

$$\hat{\tau}(\mathbf{x}) = h(\mathbf{x}, 1) - h(\mathbf{x}, 0). \tag{3}$$

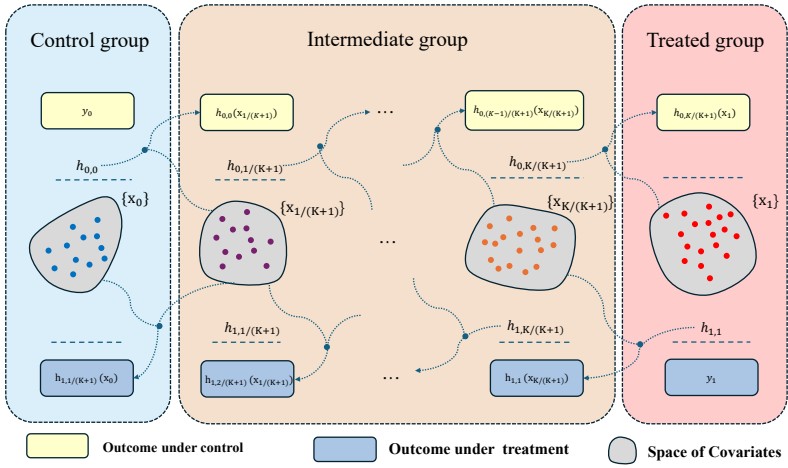

Figure 1: Overview of G-learner. In the figure, the yellow and blue boxes represent the outcomes of the corresponding groups. Here, $y_0$ and $y_1$ denote the observed factual outcomes of the control group and treated group, respectively, while the remaining outcomes are pseudo outcomes generated by $h_{0,\kappa^-}$ or $h_{1,\kappa^+}$. The observed covariates of the control and treated groups are also shown as $\mathbf{x}_0$ and $\mathbf{x}_1$; the remaining covariates $x_\kappa$ are generated via optimal transport.

For convenience, we define $f(\mathbf{x}, t) = f_t(\mathbf{x})$ and $h_t(\mathbf{x}) = h(\mathbf{x}, t)$.

Throughout the paper, $\mathbf{1}_n \in \mathbb{R}^n$ is a vector with all the entry being 1. For the matrix $\mathbf{A}$, $A_{ij}$ is the $(i, j)$-th entry, $\mathbf{A}^\top$ is the transpose of $\mathbf{A}$. $\|\mathbf{A}\|_0$ is the $\ell_0$-norm that counts the non-zero entries of $\mathbf{A}$. The probability simplex $\Sigma_n$ is defined as $\Sigma_n = \{\mathbf{v} \in \mathbb{R}^n \mid \sum_{i=1}^n v_i = 1, v_i \geq 0 \ \forall i\}$.

## 4 METHODOLOGY

Figure 1 illustrates the main idea of our proposed method G-learner. Given the control group ($t = 0$) and the treated group ($t = 1$) with covariate shift, we generate intermediate groups indexed by $\kappa \in (0, 1)$ between the control and treated groups based on optimal transport. After that, we gradually adjust outcome prediction models between groups with the help of generated data, during which the models trained on filtered samples of high quality. In the following, we present the technical details of our method.

### 4.1 INTERMEDIATE GROUP GENERATION

Due to the existence of confounders, the control and treated groups follow different distributions. As a result, the outcome prediction model trained on one group cannot perform well on the opposite group, suffering from limited generalization ability (Shalit et al., 2017). To address this, we propose to connect the control and treated groups by generating a series of intermediate groups, which is derived by the Wasserstein geodesic from the optimal transport theory.

Optimal transport seeks to find the optimal plan to move mass from one distribution to another with a minimized transport cost (Villani, 2008; Peyré et al., 2019). Formally, given the feature space $\mathcal{X}$, let $\mathcal{M}(\mathcal{X})$ be the set of Radom measures, $\alpha \in \mathcal{M}(\mathcal{X})$ and $\beta \in \mathcal{M}(\mathcal{X})$ be two distributions, whose corresponding samples are $a$ and $b$, respectively. $\gamma \in \Gamma(\alpha, \beta)$ is a transport plan, and $\Gamma(\alpha, \beta)$ is the set of all joint probability couplings whose marginal distributions are $\alpha$ and $\beta$, respectively. $\gamma(a, b)$ indicates how many masses are transported from the sample $a$ to the sample $b$, and the transport cost between them is measured by a cost function $c(a, b)$, where $c : \mathcal{X} \times \mathcal{X} \to \mathbb{R}^+$ can be implemented by a distance metric. Optimal transport minimizes the total transport cost between the distributions and defines the following $p$-Wasserstein distance

$$\mathcal{W}_p(\alpha, \beta) = \left( \inf_{\gamma \in \Gamma(\alpha, \beta)} \int_{\mathcal{X} \times \mathcal{X}} c(a, b)^p d\gamma(a, b) \right)^{\frac{1}{p}}. \tag{4}$$

In the remainder, we adopt $p = 2$ and implement the cost function by the Euclidean distance, *i.e.*, $c(a, b) = \|a - b\|_2$, which derives the 2-Wasserstein distance $\mathcal{W}_2$.

The Wasserstein distance not only measures the discrepancy between two data distributions, but also induces a geodesic structure of the space of probability measures. The geodesic path can be naturally constructed via optimal transport-based interpolation between two distributions (Villani, 2008), and the distributions along the geodesic path provide a smooth connection between two distributions. Mathematically, the distributions on the Wasserstein geodesic can be obtained by solving the following problem parameterized by $\kappa \in [0, 1]$

$$\mu_\kappa = \arg\min_\mu (1 - \kappa)\mathcal{W}_2(\alpha, \mu)^2 + \kappa\mathcal{W}_2(\beta, \mu)^2. \tag{5}$$

The constructed distribution $\mu_\kappa$ is also known as the Wasserstein barycenter of two distributions with the parameter $\kappa$, which is the mean of two probability measures under the optimal transport metric (Bonneel et al., 2015). The barycentric projection considers the special case with $\kappa = 1$ Courty et al. (2016), which obtains the representations of samples in $\mu_0$ by weighted samples in the distribution $\mu_1$.

Our target is to adjust the outcome prediction model trained on the control group to the treated group and vice versa, so that the outcome prediction models $h(\mathbf{x}, 1)$ and $h(\mathbf{x}, 0)$ can perform well on both groups. To this end, we establish an optimal transport problem between the control and treated groups, and generate the intermediate groups between them based on the Wasserstein geodesic. Through the generated intermediate groups, we can gradually adjust the outcome prediction models from one group to another, which avoids the over-balancing issue in balanced representation learning and achieves prediction models with promising generalization ability.

Specifically, for the control samples $\{\mathbf{x}_{0,i}\}_{i=1}^{n_0}$ and the treated samples $\{\mathbf{x}_{1,j}\}_{j=1}^{n_1}$, let $\{p_{0,i}\}_{i=1}^{n_0}$ and $\{p_{1,j}\}_{j=1}^{n_1}$ be the corresponding probability mass, and $\delta(\mathbf{x})$ be the Dirac function at the location $\mathbf{x}$. The empirical distributions of the control and treated groups can be represented as

$$\mu_0 = \sum_{i=1}^{n_0} p_{0,i}\delta(\mathbf{x}_{0,i}), \quad \mu_1 = \sum_{j=1}^{n_1} p_{1,j}\delta(\mathbf{x}_{1,j}), \tag{6}$$

where the first subscript $0/1$ indicates the group index while the second subscript corresponds to the sample index. The probability masses can be denoted as a probability simplex, *i.e.*, $\mathbf{p}_0 = [p_{0,1}, \ldots, p_{0,n_0}]^\top \in \Sigma_{n_0}, \mathbf{p}_1 = [p_{1,1}, \ldots, p_{1,n_1}]^\top \in \Sigma_{n_1}$. =Without additional prior information of the underlying distribution, we usually implement $\mathbf{p}_0$ and $\mathbf{p}_1$ as uniform distributions (Courty et al., 2016), *i.e.*, $p_{0,i} = \frac{1}{n_0} \forall i, p_{1,j} = \frac{1}{n_1} \forall j$. The uniform distribution can be derived from the principle of maximum entropy, which states that the probability distribution that best represents the current observations is the one with the largest entropy. The equal weights in the uniform distribution also reflect that each sample has the same importance. We also try to obtain marginal distributions based on kernel density estimation, and report the results in Appendix B.8. Based on this, the transport plan $\gamma$ is defined in the following set

$$\Gamma(\mu_0, \mu_1) = \{\gamma \in (\mathbb{R}^+)^{n_0 \times n_1} \mid \gamma\mathbf{1}_{n_1} = \mathbf{p}_0, \gamma^\top\mathbf{1}_{n_0} = \mathbf{p}_1\}, \tag{7}$$

and the optimal transport plan is obtained by solving the following optimization problem, which is the discrete form of Problem (4):

$$\gamma^* = \arg\min_\gamma \sum_{i=1}^{n_0} \sum_{j=1}^{n_1} \gamma_{ij}\|\mathbf{x}_{0,i} - \mathbf{x}_{1,j}\|_2^2 \quad \text{s.t.} \quad \gamma \in \Gamma(\mu_0, \mu_1). \tag{8}$$

This is a constrained linear programming problem that can be efficiently solved by the Earth Mover Distance solver (Flamary et al., 2021). Optimal transport requires a well-defined cost function, which is implemented by the squared Euclidean distance here. For complex types such as discrete or categorical data, one can define a distance or learn a mapping function to project them into a representation space, so that optimal transport can be applied.

Following Remark 7.1 in (Peyré et al., 2019) and Theorem 5.27 in (Santambrogio, 2015), given an optimal coupling $\pi$, the barycenter is obtained by the push-forward interpolation $\mu_\kappa = P_{\kappa,\sharp}\pi$ where $P_t : (x_0, x_1) \in \mathbb{R}^d \times \mathbb{R}^d \mapsto (1 - \kappa)x_0 + \kappa x_1$. For the discrete setup, once the optimal transport plan

$\gamma^*$ is obtained, the intermediate group derived by Eq. (5) boils down to the interpolation defined as the following empirical distribution (Villani, 2008; He et al., 2024)

$$\mu_\kappa = \sum_{i=1}^{n_0} \sum_{j=1}^{n_1} \gamma_{ij}^* \delta((1-\kappa)\mathbf{x}_{0,i} + \kappa\mathbf{x}_{1,j}). \tag{9}$$

We generate a sequence of intermediate groups with different values of $\kappa$ to connect the control and treated groups. For a given $\kappa \in [0,1]$, $\mu_\kappa$ involves $\|\gamma^*\|_0$ generated samples, in which each non-zero entry $\gamma_{ij}^* > 0$ is the probability mass of the corresponding sample. As a result, we obtain weighted generated samples $\{(\mathbf{x}_{\kappa,ij}, \gamma_{ij}^*) : \gamma_{ij}^* > 0\}$ that can be used to gradually adjust outcome models by self-training. According to (Peyré et al., 2019), the total sum of $n_\kappa$ for the set $\mathbf{x}_\kappa$ is $n_0 + n_1 - 1$.

## 4.2 PREDICTION MODEL ADJUSTMENT

To smoothly connect $\mu_0$ and $\mu_1$, we generate $K$ intermediate groups indexed by $\kappa \in \{\frac{1}{K+1}, \dots, \frac{K}{K+1}\}$, and then adjust the outcome prediction models between two consecutive groups by a self-training paradigm (He et al., 2024). In the following, we take the model $h_0(\cdot)$ predicting the potential outcome of non-treatment as the example to present our method of model adjustment. The adjustment for the model $h_1(\cdot)$ to predict the treated potential outcome is similar.

Specifically, we transfer $h_0(\cdot)$ from the group $\kappa^- = \kappa - \frac{1}{K+1}$ to the group $\kappa$, and define $h_{0,\kappa^-}(\cdot)$ as the non-treatment outcome prediction model have been trained on the group $\kappa^-$. At the initial step with $\kappa^- = 0$, the model $h_{0,0}(\cdot)$ is trained on the control group $\mu_0$ with ground-truth non-treatment factual outcomes. Although $h_{0,0}(\cdot)$ cannot perform well on the treated group $\mu_1$, it is expected to perform much better on the distribution $\mu_{1/(K+1)}$ compared with on the distribution $\mu_1$, since the covariate shift between $\mu_0$ and $\mu_{1/(K+1)}$ is much smaller than that between $\mu_0$ and $\mu_1$. Therefore, $h_{0,0}(\mathbf{x})$ can be used as supervised information for the sample $\mathbf{x} \in \mu_{1/(K+1)}$ to train $h_{0,1/(K+1)}(\cdot)$, which can be leveraged as the annotation function to train the model $h_{0,2/(K+1)}(\cdot)$ in the next step. Finally, we can obtain the prediction model $h_{0,1}(\cdot)$ to predict non-treatment potential outcomes for the treated group $\mu_1$. $h_{0,1}(\cdot)$ is gradually adjusted from the model $h_{0,0}(\cdot)$ through the intermediate groups derived from the Wasserstein geodesic.

Formally, model adjustment from the group $\kappa^-$ to the group $\kappa$ can be achieved by solving the following problem

$$h_{0,\kappa} = \arg\min_h \sum_{\mathbf{x}_\kappa \in \mu_\kappa} \ell(h(\mathbf{x}_\kappa), h_{0,\kappa^-}(\mathbf{x}_\kappa)), \tag{10}$$

where the sample $\mathbf{x}_\kappa$ in the distribution $\mu_\kappa$ is annotated by the model $h_{0,\kappa^-}(\cdot)$, and then is used to train the model $h(\cdot)$. The loss function is implemented by the squared loss based on the sample weight obtained from optimal transport, *i.e.*,

$$\ell(h(\mathbf{x}_{\kappa,ij}), h_{0,\kappa^-}(\mathbf{x}_{\kappa,ij})) = \gamma_{ij}^*(h(\mathbf{x}_{\kappa,ij}) - h_{0,\kappa^-}(\mathbf{x}_{\kappa,ij}))^2, \tag{11}$$

where $\mathbf{x}_{\kappa,ij}$ is a sample in the distribution $\mu_\kappa$ with the weight $\gamma_{ij}^*$, which is obtained according to Eq. (9).

To improve the performance of potential outcome prediction, we further refine our model by filtering generated data, which is described by the following.

### 4.2.1 GENERATED DATA FILTERING.

During the procedure of adjusting $h_0(\cdot)$ from the distribution $\mu_{\kappa^-}$ to $\mu_\kappa$, we train the model on the samples $\mathbf{x}_\kappa \in \mu_\kappa$ with the pseudo labels $h_{0,\kappa^-}(\mathbf{x}_\kappa)$. In order to leverage high-quality samples to train the model, we filter generated data to remove samples with low prediction certainty. Specifically, for the sample $\mathbf{x}_\kappa$, we conduct $M$ times dropout-enable forward passes to obtain $M$ predicted outcomes, which are denoted as $\{\hat{Y}_{0,i}(\mathbf{x}_\kappa)\}_{i=1}^M$. After that, we calculate the standard deviation of the multiple predicted results $\sigma(\mathbf{x}_\kappa)$ to measure the uncertainty of prediction. To improve reliability, we select the $r$ proportion of the generated samples with the lowest standard deviations, which enjoy high prediction certainty and are beneficial for model training (Gal & Ghahramani, 2016). As

a result, we construct a filtered group $\tilde{\mu}_\kappa$ with high quality generated samples, and refine Problem (10) as follows

$$h_{0,\kappa} = \arg\min_h \sum_{\mathbf{x}_\kappa \in \tilde{\mu}_\kappa} \ell(h(\mathbf{x}_\kappa), h_{0,\kappa^-}(\mathbf{x}_\kappa)). \tag{12}$$

Obeying a similar approach above, the model to predict treated potential outcomes $h_1(\cdot)$ is adjusted from the group $\kappa^+ = \kappa + \frac{1}{K+1}$ to the group $\kappa$ by the following

$$h_{1,\kappa} = \arg\min_h \sum_{\mathbf{x}_\kappa \in \tilde{\mu}_\kappa} \ell(h(\mathbf{x}_\kappa), h_{1,\kappa^+}(\mathbf{x}_\kappa)), \tag{13}$$

where $h_{1,1}(\cdot)$ is trained on the distribution $\mu_1$ with ground-truth treated factual outcomes, and $h_{1,0}(\cdot)$ is gradually adjusted from $h_{1,1}(\cdot)$ through the intermediate groups.

We summarize our proposed method G-learner in Algorithm 1 in the appendix.

## 4.3 THEORETICAL ANALYSIS

In this part, we analyze the estimation errors of outcomes and effects of our method. We define $\tilde{K} = K + 1$ for notational simplicity, where $K$ is the number of intermediate groups. Let $\Delta$ be the average Wasserstein distance between two consecutive distributions, $\mathcal{E}(h_{0,1}) = \int_{\mathcal{X}} (h_{0,1}(\mathbf{x}) - f_0(\mathbf{x}))^2 p(\mathbf{x}|t=1) d\mathbf{x}$ be the expected loss of the prediction model $h_{0,1}$ on the treated group, Similarly, $\mathcal{E}(h_{0,0}) = \int_{\mathcal{X}} (h_{0,0}(\mathbf{x}) - f_0(\mathbf{x}))^2 p(\mathbf{x}|t=0) d\mathbf{x}$ is the expected loss of the prediction model $h_{0,0}$ on the control group. We have the following results

**Lemma 1.** *Following (He et al., 2024), consider two arbitrary measures from group $\kappa$ and $\kappa^-$, the error of $|\mathcal{E}(h_{0,\kappa}) - \mathcal{E}(h_{0,\kappa^-})|$ is upper bounded with probability at least $1 - r$ as:*

$$|\mathcal{E}(h_{0,\kappa}) - \mathcal{E}(h_{0,\kappa^-})| \leq \mathcal{O}(\mathcal{W}_p(\mu_\kappa, \mu_{\kappa^-}) + \frac{\rho C + \sqrt{\log(1/r)}}{\sqrt{n}}) \tag{14}$$

Under Lemma 1, it can be readily observed that the error between group $\kappa$ and $\kappa^-$, as measured by the Wasserstein distance $\mathcal{W}_p(\mu_\kappa, \mu_{\kappa^-})$, depends on the sample size $n$ and the probability $r$.

**Lemma 2.** *For a filtering ratio $r \in (0,1)$, the loss $\mathcal{E}(h_{0,1})$ is upper bounded with probability at least $1 - r$ as*

$$\mathcal{E}(h_{0,1}) \leq \mathcal{E}(h_{0,0}) + \mathcal{O}\left(\tilde{K}\Delta + \frac{\tilde{K}}{\sqrt{n}} + \tilde{K}\sqrt{\frac{\log(1/r)}{n}} + \frac{1}{\sqrt{n\tilde{K}}} + \sqrt{\frac{(\log n\tilde{K})^{3L-2}}{n\tilde{K}}} + \sqrt{\frac{\log(1/r)}{n\tilde{K}}}\right), \tag{15}$$

*where $n$ is the number of training samples, $L$ is the model depth, and $\tilde{K}\Delta$ is the accumulated distribution shift between the control and treated groups through intermediate groups.*

In addition, let $\mathcal{E}(h_{1,1}) = \int_{\mathcal{X}} (h_{1,1}(\mathbf{x}) - f_1(\mathbf{x}))^2 p(\mathbf{x}|t=1) d\mathbf{x}$ is the expected loss of the prediction model $h_{1,1}$ on the treated group. The effect estimation error is measured by the pairwise precision in the estimation of heterogeneous effect (PEHE) is defined as $\epsilon_{PEHE} = \int_{\mathcal{X}} (\hat{\tau}(\mathbf{x}) - \tau(\mathbf{x}))^2 p(\mathbf{x}) d\mathbf{x}$ (Shalit et al., 2017). According to Lemma 2, the effect estimation error is upper bounded by the following theorem.

**Theorem 1.** *The effect estimation error $\epsilon_{PEHE}$ is upper bounded by:*

$$\epsilon_{PEHE} \leq 2\mathcal{E}(h_{0,0}) + 2\mathcal{E}(h_{1,1}) + 2\mathcal{O}\left(\tilde{K}\Delta + \mathcal{B}(n, \tilde{K}, L, r)\right), \tag{16}$$

*where*

$$\mathcal{B}(n, \tilde{K}, L, r) = \frac{\tilde{K}}{\sqrt{n}} + \tilde{K}\sqrt{\frac{\log(1/r)}{n}} + \frac{1}{\sqrt{n\tilde{K}}} + \sqrt{\frac{(\log n\tilde{K})^{3L-2}}{n\tilde{K}}} + \sqrt{\frac{\log(1/r)}{n\tilde{K}}}. \tag{17}$$

The proof of Lemma 2 and Theorem 1 is given in the appendix. Theorem 1 indicates that the $\epsilon_{PEHE}$ is upper bound by the prediction error of the treated and control group, with an additional term depending on $n, \tilde{K}, r$ and $L$. In particular, with fixed $\tilde{K}, r$, and $L$, the additional term approaches zero as $n \to \infty$.

Table 1: Results for out-sample performance on real-world datasets in terms of mean and standard deviation. A lower metric indicates better performance. We highlight the best results in bold and underline the second best results.

| Method | News | | Twins | | Jobs | |
|---|---|---|---|---|---|---|
| | $\sqrt{\epsilon_{PEHE}}$ | $\epsilon_{ATE}$ | $\sqrt{\epsilon_{PEHE}}$ | $\epsilon_{ATE}$ | $R_{POL}$ | $\hat{\epsilon}_{ATT}$ |
| OLS | $4.0051 \pm 1.6369$ | $0.4777 \pm 0.2632$ | $0.5212 \pm 0.4070$ | $0.0128 \pm 0.0150$ | $0.2612 \pm 0.0632$ | $0.1825 \pm 0.1224$ |
| BART | $7.6180 \pm 2.2501$ | $5.8950 \pm 1.6207$ | $0.3246 \pm 0.0082$ | $0.0305 \pm 0.0075$ | $0.2770 \pm 0.0603$ | $0.1133 \pm 0.1092$ |
| T-learner | $2.9004 \pm 0.9077$ | $0.5526 \pm 0.4227$ | $0.3385 \pm 0.0087$ | $0.0224 \pm 0.0126$ | $0.2769 \pm 0.0180$ | $0.1409 \pm 0.0962$ |
| k-NN | $13.0410 \pm 11.2283$ | $9.5025 \pm 5.2287$ | $0.3735 \pm 0.0087$ | $0.0330 \pm 0.0089$ | $0.2642 \pm 0.0686$ | $0.1731 \pm 0.1494$ |
| PSM | $7.0947 \pm 2.1623$ | $5.3695 \pm 1.4663$ | $0.3954 \pm 0.0072$ | $0.0168 \pm 0.0047$ | $0.2667 \pm 0.0746$ | $0.1495 \pm 0.1190$ |
| GANITE | $3.7838 \pm 1.2759$ | $1.7325 \pm 0.6216$ | $\underline{0.3202 \pm 0.0085}$ | $0.0130 \pm 0.0055$ | $0.2811 \pm 0.1308$ | $0.1648 \pm 0.1083$ |
| DKLite | $3.4381 \pm 1.3630$ | $0.8782 \pm 0.5693$ | $0.3207 \pm 0.0086$ | $\underline{0.0086 \pm 0.0041}$ | $0.1730 \pm 0.0001$ | $0.1470 \pm 0.1059$ |
| TARNet | $2.1524 \pm 0.6421$ | $4.2959 \pm 0.3899$ | $0.3413 \pm 0.0085$ | $0.0129 \pm 0.0054$ | $0.2364 \pm 0.0672$ | $0.0920 \pm 0.0810$ |
| DragonNet | $3.2137 \pm 0.8048$ | $0.7673 \pm 0.5775$ | $0.4454 \pm 0.0184$ | $0.0110 \pm 0.0101$ | $\mathbf{0.0957 \pm 0.0810}$ | $0.2516 \pm 0.0506$ |
| BNN | $4.2182 \pm 1.2550$ | $2.4453 \pm 0.6801$ | $0.3202 \pm 0.0085$ | $0.0132 \pm 0.0046$ | $0.2576 \pm 0.1275$ | $\underline{0.0782 \pm 0.0755}$ |
| $CFR_{Wass}$ | $3.0044 \pm 1.0921$ | $1.0049 \pm 0.7090$ | $0.3217 \pm 0.0095$ | $0.0220 \pm 0.0254$ | $0.2393 \pm 0.0703$ | $0.0920 \pm 0.0808$ |
| $CFR_{MMD}$ | $3.0688 \pm 1.2633$ | $1.0373 \pm 0.7943$ | $0.3233 \pm 0.0085$ | $0.0279 \pm 0.0185$ | $0.2390 \pm 0.0760$ | $0.0938 \pm 0.0813$ |
| CFR-Pro | $3.3837 \pm 1.2978$ | $0.5963 \pm 0.4602$ | $0.3206 \pm 0.0088$ | $0.0161 \pm 0.0133$ | $0.2935 \pm 0.0623$ | $0.0937 \pm 0.0863$ |
| CE-RCFR | $3.2956 \pm 1.0875$ | $1.2729 \pm 0.4131$ | $0.3214 \pm 0.0095$ | $0.0244 \pm 0.0202$ | $0.1781 \pm 0.0401$ | $0.1265 \pm 0.1042$ |
| DESCN | $4.3308 \pm 1.2460$ | $2.6741 \pm 0.6953$ | $0.3244 \pm 0.0081$ | $0.0168 \pm 0.0217$ | $0.3011 \pm 0.0509$ | $0.1920 \pm 0.0078$ |
| ESCFR | $\mathbf{2.7435 \pm 0.9110}$ | $\underline{0.4255 \pm 0.3082}$ | $0.3209 \pm 0.0085$ | $0.0147 \pm 0.0083$ | $0.2396 \pm 0.0438$ | $0.0893 \pm 0.0801$ |
| G-learner | $\underline{2.8681 \pm 0.8696}$ | $\mathbf{0.2451 \pm 0.1955}$ | $\mathbf{0.3200 \pm 0.0086}$ | $\mathbf{0.0084 \pm 0.0060}$ | $\underline{0.1691 \pm 0.0622}$ | $\mathbf{0.0596 \pm 0.0740}$ |

Table 2: Results on out-sample simulated dataset in terms of mean and standard deviation. A lower metric indicates better performance and we highlight the best results in bold and underline the second best results.

| Method | $m_c = 0.5$ | | $m_c = 0.8$ | | $m_c = 1.1$ | | $m_c = 1.4$ | |
|---|---|---|---|---|---|---|---|---|
| | $\sqrt{\epsilon_{PEHE}}$ | $\epsilon_{ATE}$ | $\sqrt{\epsilon_{PEHE}}$ | $\epsilon_{ATE}$ | $\sqrt{\epsilon_{PEHE}}$ | $\epsilon_{ATE}$ | $\sqrt{\epsilon_{PEHE}}$ | $\epsilon_{ATE}$ |
| OLS | $1.12 \pm 0.50$ | $0.30 \pm 0.22$ | $1.13 \pm 0.50$ | $0.30 \pm 0.22$ | $1.16 \pm 0.50$ | $0.30 \pm 0.24$ | $1.31 \pm 0.59$ | $0.35 \pm 0.28$ |
| BART | $4.07 \pm 0.66$ | $0.98 \pm 0.31$ | $4.03 \pm 0.59$ | $0.41 \pm 0.28$ | $4.28 \pm 0.53$ | $0.45 \pm 0.27$ | $4.71 \pm 0.50$ | $1.14 \pm 0.37$ |
| T-learner | $8.23 \pm 0.78$ | $0.41 \pm 0.34$ | $8.93 \pm 0.78$ | $2.79 \pm 0.51$ | $10.20 \pm 0.70$ | $5.40 \pm 0.50$ | $12.10 \pm 0.63$ | $8.07 \pm 0.53$ |
| kNN | $4.49 \pm 0.84$ | $1.32 \pm 0.22$ | $3.76 \pm 0.91$ | $0.85 \pm 0.24$ | $3.69 \pm 0.71$ | $0.37 \pm 0.22$ | $4.19 \pm 0.56$ | $0.21 \pm 0.11$ |
| PSM | $1.98 \pm 0.59$ | $0.51 \pm 0.53$ | $2.68 \pm 0.69$ | $0.61 \pm 0.41$ | $4.16 \pm 0.99$ | $0.86 \pm 0.61$ | $5.76 \pm 1.24$ | $1.96 \pm 0.88$ |
| GANITE | $0.85 \pm 0.08$ | $0.28 \pm 0.22$ | $1.49 \pm 0.03$ | $1.47 \pm 0.06$ | $1.50 \pm 0.01$ | $1.50 \pm 0.01$ | $1.51 \pm 0.01$ | $1.50 \pm 0.01$ |
| DKLite | $0.59 \pm 0.11$ | $\underline{0.07 \pm 0.05}$ | $2.73 \pm 0.69$ | $2.47 \pm 0.78$ | $5.84 \pm 0.66$ | $5.73 \pm 0.65$ | $8.96 \pm 0.63$ | $8.89 \pm 0.64$ |
| TARNet | $0.55 \pm 0.05$ | $0.53 \pm 0.05$ | $0.55 \pm 0.04$ | $0.53 \pm 0.05$ | $0.54 \pm 0.06$ | $0.52 \pm 0.06$ | $0.55 \pm 0.05$ | $0.53 \pm 0.05$ |
| DragonNet | $\underline{0.27 \pm 0.04}$ | $0.05 \pm 0.03$ | $0.49 \pm 0.25$ | $0.05 \pm 0.05$ | $\underline{0.38 \pm 0.15}$ | $\mathbf{0.06 \pm 0.05}$ | $0.46 \pm 0.22$ | $\underline{0.10 \pm 0.09}$ |
| BNN | $0.42 \pm 0.01$ | $0.40 \pm 0.01$ | $\underline{0.48 \pm 0.01}$ | $0.46 \pm 0.01$ | $0.44 \pm 0.01$ | $0.41 \pm 0.01$ | $\underline{0.44 \pm 0.01}$ | $0.42 \pm 0.01$ |
| $CFR_{Wass}$ | $0.52 \pm 0.03$ | $0.50 \pm 0.03$ | $0.53 \pm 0.05$ | $0.51 \pm 0.05$ | $0.54 \pm 0.07$ | $0.52 \pm 0.07$ | $0.55 \pm 0.05$ | $0.53 \pm 0.05$ |
| $CFR_{MMD}$ | $0.51 \pm 0.03$ | $0.49 \pm 0.03$ | $0.51 \pm 0.03$ | $0.49 \pm 0.03$ | $0.50 \pm 0.08$ | $0.48 \pm 0.08$ | $0.96 \pm 1.38$ | $0.56 \pm 0.27$ |
| CFR-Pro | $0.54 \pm 0.41$ | $0.53 \pm 0.41$ | $0.56 \pm 0.43$ | $0.55 \pm 0.45$ | $0.56 \pm 0.43$ | $0.55 \pm 0.45$ | $0.75 \pm 0.14$ | $0.46 \pm 0.10$ |
| CE-RCFR | $0.47 \pm 0.02$ | $0.46 \pm 0.03$ | $0.50 \pm 0.03$ | $0.49 \pm 0.03$ | $0.52 \pm 0.02$ | $0.50 \pm 0.01$ | $0.52 \pm 0.02$ | $0.49 \pm 0.02$ |
| DESCN | $0.52 \pm 0.05$ | $0.49 \pm 0.06$ | $0.52 \pm 0.05$ | $0.49 \pm 0.06$ | $0.51 \pm 0.06$ | $0.49 \pm 0.01$ | $0.52 \pm 0.01$ | $0.49 \pm 0.01$ |
| ESCFR | $0.59 \pm 0.15$ | $0.12 \pm 0.07$ | $0.75 \pm 0.18$ | $\underline{0.24 \pm 0.11}$ | $0.96 \pm 0.21$ | $0.40 \pm 0.12$ | $0.96 \pm 0.21$ | $0.40 \pm 0.12$ |
| G-learner | $\mathbf{0.20 \pm 0.03}$ | $\mathbf{0.05 \pm 0.04}$ | $\mathbf{0.23 \pm 0.04}$ | $\mathbf{0.05 \pm 0.05}$ | $\mathbf{0.29 \pm 0.07}$ | $\underline{0.08 \pm 0.05}$ | $\mathbf{0.36 \pm 0.14}$ | $\mathbf{0.09 \pm 0.11}$ |

## 5 EXPERIMENTS

In this section, we first describe the experimental settings, including the compared methods and evaluation metrics. After that, we present experimental results and discussions on real-word and simulation datasets.

### 5.1 EXPERIMENTAL SETTINGS

#### 5.1.1 COMPARED METHODS.

We compare the performance of G-learner with classical and state-of-the-art methods, including **OLS**, **BART** (Chipman et al., 2010; Hill, 2011), **PSM** (Rosenbaum & Rubin, 1983), $k$-**NN** (Crump et al., 2008), **T-learner** (Künzel et al., 2019), **TARNet** (Shalit et al., 2017), **BNN** (Johansson et al., 2016), **CFR**$_{Wass}$ and **CFR**$_{MMD}$ (Shalit et al., 2017), **GANITE** (Yoon et al., 2018), **DragonNet** (Shi et al., 2019), **DKLite** (Zhang et al., 2020), **CFR-Pro** (Wang et al., 2025a), **CE-RCFR** (Wang et al., 2024a), **DESCN** (Zhong et al., 2022), and **ESCFR** (Wang et al., 2023).

### 5.1.2 EVALUATION METRICS.

Following (Shalit et al., 2017), we evaluate the performance of different methods based on the Precision in Estimation of Heterogeneous Effect error ($\epsilon_{PEHE}$) and the absolute error in estimating the Average Treatment Effect ($\epsilon_{ATE}$). Additionally, we also assess performance using the policy risk ($R_{POL}$) and the error in the Average Treatment Effect on the Treated group ($\epsilon_{ATT}$). Complete formulas of these metrics are given in the Appendix.

## 5.2 EXPERIMENTS ON REAL-WORLD DATA

We compare the performance of different methods on three real-world datasets: News, Twins, and Jobs. (i) The News dataset includes 5,000 news articles, which are represented by embedding each article in a topic space and defining two device centroids (desktop and mobile). (Johansson et al., 2016) (ii) The Twins dataset is collected from U.S. twins born in 1989–1991 (Almond et al., 2005) For each twin pair, we record both treatment conditions, including lighter ($t = 0$) and heavier ($t = 1$), and the one-year mortality outcome. To simulate the confounding bias, we choose one of the twins as follows: $t \sim \text{Bernoulli}\big(\sigma(\mathbf{w}^\top \mathbf{x} + b)\big)$, where $\mathbf{w}$ is drawn uniformly from $(-0.1, 1)^{30}$ and $b$ from $\mathcal{N}(0, 0.1)$, and $\sigma$ is the sigmoid function. (iii) The Jobs dataset includes experimental samples (297 treated samples and 425 control samples) and the *Panel Study of Income Dynamics* (PSID) comparison group (2490 control), which not only capitalizes on the internal validity of the Randomized Controlled Trial (RCT) but also draws on large-scale observational data to boost inferential precision and enhance generalizability (LaLonde, 1986). Following the experiments in (Johansson et al., 2016; Machlanski et al., 2023), the News and Twins frameworks provide both factual and counterfactual outcomes for two-treatment causal inference, so we assess methods using $\epsilon_{PEHE}$ and $\epsilon_{ATE}$. In contrast, Jobs supplies only factual outcomes, which we evaluate via $\hat{R}_{POL}$ and $\epsilon_{ATT}$. More details of the datasets can be found in the appendix.

Table 1 shows the results of different methods on the real-world datasets under the out-sample setting. The results under the in-sample setting are shown in the appendix. We draw several interesting observations as follows.

- Overall, our proposed method achieves best or highly competitive performance on all the datasets in terms of both evaluation metrics, which clearly demonstrates the effectiveness and robustness of it. These results verify the practical value and generalizability of our method in different scenarios.

- Among the compared methods, OLS, BART, and T-learner train separate neural regressors for each treatment group, TARNet and DragonNet train shared representation layers and separate outcome heads. These methods suffer from the issue of covariate distribution shifts. Compared with them, G-learner demonstrates superior performance, which verifies that our method can learn effective prediction models with promising generalization ability across groups.

- Compared with the generative method GANITE, G-learner leverages optimal transport to generate intermediate data. Beneficial from the geometric information involved in data extracted by optimal transport, our method achieves better performance.

- The balanced representation learning method, such as BNN, CFR, and ESCFR, adjusts confounders to reduce the confounding bias, which suffers from the overbalancing issue and could discard discriminative features for potential outcome prediction. During the balancing procedure, some important information could be removed hampering the causal inference performance. Rather than adjusting the confounders, G-learner adjusts the outcome prediction models by gradually transferring the models between groups with the aid of intermediate groups. By doing this, G-learner leverages all the features to train the prediction model without information loss, avoiding the overbalancing issue and achieving better performance.

## 5.3 EXPERIMENTS ON SIMULATION DATA

Table 2 shows the results of different methods on simulation data under the out-sample setting. The results under the in-sample setting are shown in the appendix. We evaluate the robustness

of each method by incrementally widening the group discrepancies, thereby emulating a range of confounding biases. As the degree of confounding bias increases, all the methods show worse performance, which indicates that the distribution shift between groups affects the performance of causal inference. Nevertheless, compared to other methods, G-learner consistently achieves the best performance, demonstrating its ability to preserve robust predictive accuracy and stability even as the severity of confounding bias increases.

## 6  CONCLUSION

In this paper, we propose a model adjustment method to learn outcome prediction models that can perform well on both control and treated groups. To achieve this, we generate a series of intermediate groups between the control and treated groups through the Wasserstein geodesic, which is derived from the optimal transport theory. By doing this, the control and treated groups with significant distribution shift can be smoothly connected, and the outcome prediction model can be gradually adjusted between them. We further propose a generated data filtering method to refine our prediction model. We analyze the upper bound of the effect estimation error of our method, and conduct experiments on simulation and real-world datasets to evaluate the performance of our method. We provide an alternative approach in addition to covariate adjustment for addressing the issue of confounding bias in causal inference.

## 7  ACKNOWLEDGMENTS

This research was supported in part by National Science and Technology Major Project (2021ZD0111501), National Natural Science Foundation of China (62206061, U24A20233), National Science Fund for Excellent Young Scholars (62122022), Guangdong Basic and Applied Basic Research Foundation (2024A1515011901), Guangzhou Basic and Applied Basic Research Foundation (2023A04J1700), CCF-DiDi GAIA Collaborative Research Funds (CCF-DiDi GAIA 202521).

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

# A  PSEUDO-CODE OF G-LEARNER

Algorithm 1 presents the pseudo-code of our method G-learner.

---

**Algorithm 1** G-learner.

---

**Require:** factual samples $\{\mathbf{x}_i, t_i, y_i\}_{i=1}^n$, number of intermediate groups $K$, proportion parameter $r$.
 1: Train the model $h_{0,0}$ to predict non-treatment outcomes on the control group $\mu_0$.
 2: Train the model $h_{1,1}$ to predict treated outcomes on the treated group $\mu_1$.
 3: Obtain the optimal transport plan $\gamma^*$ by solving Problem (8).
 4: Generate $K$ intermediate groups $\{\mu_\kappa\}_{\kappa=1}^K$ according to Eq. (9).
 5: **for** $\kappa$ in $\frac{1}{K+1}, \ldots, \frac{K}{K+1}$ **do**
 6:     Select the $r$ proportion of generated samples to obtain $\tilde{\mu}_\kappa$.
 7:     Train the model $h_{0,\kappa}(\cdot)$ via Problem (12).
 8:     Select the $r$ proportion of generated samples to obtain $\tilde{\mu}_{1-\kappa}$.
 9:     Train the model $h_{1,1-\kappa}(\cdot)$ via Problem (13).
10: **end for**
11: **Output:** Outcome prediction models $h_{0,1}(\cdot)$ and $h_{1,0}(\cdot)$.

---

# B  EXPERIMENT

All the experiments are run on a single 24GB GPU of NVIDIA GeForce RTX 4090 GPU. Following, we introduce more details about the experiment, including more details of the datasets, formulations of evaluation metrics, results of the in-sample setting, and visualization of intermediate group generation.

## B.1  MORE DETAILS OF DATASETS

**News**  The News dataset is first proposed as a benchmark for counterfactual inference by (Johansson et al., 2016). The News dataset simulates counterfactual inference by modeling news articles as topic distributions $z(\mathbf{x})$, derived from a topic model trained on the NY Times corpus. Multiple centroids are randomly chosen in the topic space, where one centroid represents the control group, while the other centroids represent treated groups viewing devices (treatments). Each centroid $z_j$ is associated with a Gaussian outcome distribution: $m_j \sim \mathcal{N}(0.45, 0.15)$, $\sigma_j \sim \mathcal{N}(0.1, 0.05)$, from which ideal potential outcomes are sampled as $\tilde{y}_j \sim \mathcal{N}(m_j, \sigma_j) + \epsilon$, where $\epsilon \sim \mathcal{N}(0, 0.15)$. The unscaled potential outcomes are computed as $\bar{y}_j = \tilde{y}_j \cdot [D(z(\mathbf{x}), z_j) + D(z(\mathbf{x}), z_c)]$, where $D(\cdot, \cdot)$ is the Euclidean distance, and $z_c$ represents the control centroid. The treatment assignment follows $t|x \sim \text{Bernoulli}(\text{softmax}(\nu\bar{y}_j))$, with $\nu$ controlling the strength of assignment bias ($\nu = 0$ implies no bias). The true observed outcomes are scaled by a constant $D = 50$: $y_j = D \cdot \bar{y}_j$. The dataset can simulate $k = 2$ treatments with $\nu = 10$, enabling flexible modeling of counterfactual inference scenarios.

**Twins**  The Twins Dataset comprises twins born in the United States between 1989 and 1991, with 30 covariates related to pregnancy, birth, and parental characteristics. In causal inference studies, the treatment is defined as the heavier twin, and the outcome is one-year mortality. This dataset is widely used to assess the causal effect of birth weight differences on infant survival. (Almond et al., 2005) For each twin pair, we record both treatment conditions, including lighter ($t = 0$) and heavier ($t = 1$), and the one-year mortality outcome. To simulate the confounding bias, we choose one of the twins as follows: $t \sim \text{Bernoulli}(\sigma(\mathbf{w}^\top \mathbf{x} + b))$, where $w$ is drawn uniformly from $(-0.1, 1)^{30}$ and $b$ from $\mathcal{N}(0, 0.1)$, and $\sigma$ is the sigmoid function.

**Jobs**  The study by (LaLonde, 1986) is a widely used benchmark in the causal inference community, where the treatment is job training and the outcomes are income and employment status after training. 1. This dataset combines a randomized study based on the National Supported Work program with observational data to form a larger dataset (Smith & Todd, 2005). The presence of the randomized subgroup gives a way to estimate the "ground truth" causal effect. The study includes 8

covariates, such as age and education, as well as previous earnings. We construct a binary classification task, called Jobs, where the goal is to predict unemployment, using the feature set of (Dehejia & Wahba, 2002). Following (Smith & Todd, 2005), we use the LaLonde experimental sample (297 treated, 425 control) and the PSID 2. This dataset combines a randomized study based on the National Supported Work program with observational data to form a larger dataset (Smith & Todd, 2005). The presence of the randomized subgroup gives a way to estimate the "ground truth" causal effect. The study includes 8 covariates such as age and education, as well as previous earnings. We construct a binary classification task, called Jobs, where the goal is to predict unemployment, using the feature set of (Dehejia & Wahba, 2002). Following (Smith & Todd, 2005), we use the LaLonde experimental sample (297 treated, 425 control) and the PSID comparison group (2490 control). There were 482 (15%) subjects unemployed by the end of the study.

**Simulation**  We generate 1,500 treated samples from a multivariate normal distribution $\mathbf{x}_t \sim \mathcal{N}(m_t^{10\times1}, 0.5\,\Sigma_t\Sigma_t^\top)$, and 1,500 control samples from $\mathbf{x}_c \sim \mathcal{N}(m_c^{10\times1}, 0.5\,\Sigma_c\Sigma_c^\top)$, where each $\Sigma$ is drawn from a uniform distribution $\Sigma_{\cdot} \sim U\big((0,\,m_{\cdot})^{10\times10}\big)$. We fix $m_t = 0.5$ and vary $m_c$ to simulate different levels of confounding bias. The potential outcomes are generated as $Y_1 = 0.5 + \mathbf{w}^\top\mathbf{x} + \xi$ and $Y_0 = \mathbf{w}^\top\mathbf{x} + \xi$, where the weight vector $\mathbf{w}$ is drawn from $\mathbf{w} \sim \mathcal{N}(\mathbf{1}, 0.01^2\,I^{10}), \mathbf{w} \in \mathbb{R}^{10}$, $\mathbf{1}$ denotes a length-10 vector of ones, and $I^{10}$ is the $10 \times 10$ identity matrix. The noise term $\xi$ is generated as $\xi \sim \mathcal{N}(0, 0.1^2)$.

### B.2  COMPARE METHODS

Following is the details of compare methods:

- Statistics-based methods: **OLS** trains separate linear regression models for the control and treated groups to predict potential outcomes. **BART** (Chipman et al., 2010; Hill, 2011) yields posterior estimates of treatment effects, supporting uncertainty quantification.

- Matching Methods: **PSM** (Rosenbaum & Rubin, 1983) estimates treatment effects by matching treated and control units using propensity scores derived from logistic regression. $k$-**NN** (Crump et al., 2008) predicts potential outcomes using factual outcomes of the $k$ nearest neighbors in the opposite group.

- Neural Network-based methods: **T-learner** (Künzel et al., 2019) trains a separate neural network for each treatment group. **TARNet** (Shalit et al., 2017) reduces distribution imbalance by learning shared latent representations of covariates. **BNN** (Johansson et al., 2016) combines domain adaptation and representation learning to minimize the discrepancy distance in the hypothesis space. **CFR**$_{Wass}$ and **CFR**$_{MMD}$ (Shalit et al., 2017) align the distributions of treated and control groups in a latent space by minimizing the Integral Probability Metric, which is implemented by the Wasserstein distance and Maximum Mean Discrepancy (MMD), respectively. **GANITE** (Yoon et al., 2018) leverages a generative adversarial network to output counterfactual outcomes for ITE estimation. **DragonNet** (Shi et al., 2019) leverages propensity scores and targeted regularization to improve outcome prediction and stabilize treatment effect estimation. **DKLite** (Zhang et al., 2020) learns invertible representations with overlapping support and standardizes using counterfactual variance to estimate ITE. **CFR-Pro** (Wang et al., 2025a) enhances HTE estimation from observational data by incorporating local proximity via an optimal transport regularization and a dimension-reducing subspace projector. **CE-RCFR** (Wang et al., 2024a) boosts ITE estimation with relaxed optimal transport alignment and consensus aggregation, taming mini-batch noise and gradient conflict in one shot. **DESCN** (Zhong et al., 2022) jointly models treatment and response in the full sample space to curb bias and imbalance. **ES-CFR** (Wang et al., 2023) leverages optimal transport to align the distributions of treated and control groups and address the unobserved confounder issue.

Table 3: Results for in-sample performance on real-world datasets in terms of mean and standard deviation. A lower metric indicates better performance. We highlight the best results in bold and underline the second best results

.

| Method | News | | Twins | | Jobs | |
|---|---|---|---|---|---|---|
| | $\sqrt{\epsilon_{PEHE}}$ | $\epsilon_{ATE}$ | $\sqrt{\epsilon_{PEHE}}$ | $\epsilon_{ATE}$ | $R_{POL}$ | $\hat{\epsilon}_{ATT}$ |
| OLS | $4.0069 \pm 1.6867$ | $0.4698 \pm 0.2580$ | $0.4299 \pm 0.1670$ | $0.0061 \pm 0.0059$ | $0.2380 \pm 0.0427$ | $0.1015 \pm 0.0417$ |
| BART | $2.6705 \pm 0.8685$ | $0.5464 \pm 0.2877$ | $0.3208 \pm 0.0027$ | $\mathbf{0.0034 \pm 0.0028}$ | $0.2614 \pm 0.0305$ | $0.0939 \pm 0.0452$ |
| T-learner | $2.5736 \pm 0.7379$ | $0.5498 \pm 0.3293$ | $0.3335 \pm 0.0038$ | $0.0221 \pm 0.0108$ | $0.2796 \pm 0.0491$ | $0.1532 \pm 0.1382$ |
| k-NN | $7.5610 \pm 11.4923$ | $0.3561 \pm 0.5392$ | $0.3705 \pm 0.0021$ | $0.0054 \pm 0.0034$ | $0.1939 \pm 0.0242$ | $0.1256 \pm 0.1263$ |
| PSM | $2.9310 \pm 0.7843$ | $\underline{0.3333 \pm 0.2515}$ | $0.3688 \pm 0.0028$ | $\underline{0.0057 \pm 0.0044}$ | $0.2486 \pm 0.0231$ | $0.1183 \pm 0.0929$ |
| GANITE | $3.8045 \pm 1.2562$ | $1.7301 \pm 0.6204$ | $0.3193 \pm 0.0021$ | $0.0169 \pm 0.0014$ | $0.2058 \pm 0.0616$ | $0.1861 \pm 0.0804$ |
| DKLite | $3.4522 \pm 1.3588$ | $0.8766 \pm 0.5675$ | $\underline{0.3193 \pm 0.0021}$ | $0.0067 \pm 0.0041$ | $0.2870 \pm 0.0115$ | $0.1502 \pm 0.0344$ |
| TARNet | $1.4367 \pm 0.3708$ | $0.3445 \pm 0.2771$ | $\underline{0.3193 \pm 0.0021}$ | $0.0168 \pm 0.0014$ | $0.2498 \pm 0.0398$ | $0.0849 \pm 0.0250$ |
| DragonNet | $2.4641 \pm 0.7849$ | $0.8160 \pm 0.7010$ | $0.3614 \pm 0.0053$ | $0.0065 \pm 0.0064$ | $\mathbf{0.1088 \pm 0.0923}$ | $0.0782 \pm 0.0244$ |
| BNN | $4.2649 \pm 1.3215$ | $2.4577 \pm 0.7031$ | $0.3194 \pm 0.0021$ | $0.0171 \pm 0.0042$ | $0.2196 \pm 0.0433$ | $\underline{0.0714 \pm 0.0229}$ |
| $CFR_{Wass}$ | $2.0692 \pm 0.7105$ | $0.8396 \pm 0.4813$ | $0.3205 \pm 0.0040$ | $0.0196 \pm 0.0239$ | $0.2386 \pm 0.0490$ | $0.0845 \pm 0.0250$ |
| $CFR_{MMD}$ | $2.1810 \pm 0.9814$ | $0.9752 \pm 0.7153$ | $0.3224 \pm 0.0038$ | $0.0302 \pm 0.0183$ | $0.2503 \pm 0.0418$ | $0.0865 \pm 0.0253$ |
| CFR-Pro | $3.3859 \pm 1.3643$ | $0.5562 \pm 0.4728$ | $0.3197 \pm 0.0023$ | $0.0183 \pm 0.0156$ | $0.3007 \pm 0.0216$ | $0.0938 \pm 0.0184$ |
| CE-RCFR | $3.2342 \pm 1.0884$ | $1.3131 \pm 0.4016$ | $0.3204 \pm 0.0027$ | $0.0244 \pm 0.0195$ | $0.2886 \pm 0.0378$ | $0.1132 \pm 0.0524$ |
| DESCN | $4.3748 \pm 1.3147$ | $2.6976 \pm 0.7238$ | $0.3174 \pm 0.0031$ | $0.0149 \pm 0.0105$ | $0.2811 \pm 0.0349$ | $0.1620 \pm 0.0178$ |
| ESCFR | $\mathbf{1.7233 \pm 0.3845}$ | $0.3678 \pm 0.2559$ | $0.3191 \pm 0.0021$ | $0.0159 \pm 0.0085$ | $0.2976 \pm 0.0285$ | $0.0817 \pm 0.0253$ |
| G-learner | $2.2750 \pm 0.7415$ | $\mathbf{0.2258 \pm 0.1472}$ | $\mathbf{0.3191 \pm 0.0020}$ | $0.0107 \pm 0.0039$ | $0.2053 \pm 0.0151$ | $\mathbf{0.0552 \pm 0.0306}$ |

## B.3 FORMULATIONS OF EVALUATION METRICS

Following (Shalit et al., 2017), we evaluate $R_{POL}$ and $\epsilon_{ATT}$ on Jobs dataset:

$$R_{POL} = 1 - (\mathbb{E}[Y_1|\pi_h(\mathbf{x}) = 1, t = 1] \cdot p(\pi_h = 1) + \mathbb{E}[Y_0|\pi_h(\mathbf{x}) = 0, t = 0] \cdot p(\pi_h = 0)),$$

$$\epsilon_{ATT} = |(|T|^{-1} \sum_{i \in T} y_i - |C \cap E|^{-1} \sum_{i \in C \cap E} y_i) - |T|^{-1} \sum_{i \in T} (h_1(x_i) - h_0(x_i))|,$$

where for a model $h$, if $h_1(\mathbf{x}) - h_0(\mathbf{x}) > \lambda$, $\pi_h(\mathbf{x}) = 1$ else $\pi_h(\mathbf{x}) = 0$. Besides, we evaluate $\epsilon_{PEHE}$ and $\epsilon_{ATE}$ on other datasets:

$$\epsilon_{PEHE} = \frac{1}{n} \sum_{i=1}^{n} [(h_1(x_i) - h_0(x_i)) - (f_1(x_i) - f_0(x_i))]^2,$$

$$\epsilon_{ATE} = |\frac{1}{n} \sum_{i=1}^{n} (h_1(x_i) - h_0(x_i)) - \frac{1}{n} \sum_{i=1}^{n} (f_1(x_i) - f_0(x_i))|.$$

## B.4 EXTERNAL RESULT OF IN-SAMPLE SETTING

We give the results of real-world and simulation datasets under the in-sample setting in Table 3 and Table 4.

## B.5 VISUALIZATION OF INTERMEDIATE GROUP GENERATION

In this part, we conduct experiments on a toy 2D dataset to visualize the generated intermediate data. We construct a control group and a treated group with means of -1 and 1, respectively, both having a standard deviation of 0.1 and covariate dimensionality of 2. The visualization results are shown in Figure 2. In Figure 2(a), we present the original covariates separately for the control and treated groups. In Figure 2(b), eight intermediate groups are generated via optimal transport between the control and treated distributions. These intermediate nodes lie precisely along the geodesic connecting the two groups.

## B.6 SENSITIVITY ANALYSIS

In order to assess the impact of hyperparameter configurations on model performance, we take simulation data where $m_c = 1.4$ as an example to evaluate the performance of our method with varying values of hyperparameters. Figure 3(a) and Figure 3(b) show the results of different numbers of intermediate groups. We observe that when the number of intermediate groups increases from 0 to 4,

Table 4: Result on simulated dataset in terms of mean and standard deviation. A lower metric indicates better performance. We highlight the best results in bold and underline the second best results.

| Method | $m_c = 0.5$ | | $m_c = 0.8$ | | $m_c = 1.1$ | | $m_c = 1.4$ | |
|---|---|---|---|---|---|---|---|---|
| | $\sqrt{\epsilon_{PEHE}}$ | $\epsilon_{ATE}$ | $\sqrt{\epsilon_{PEHE}}$ | $\epsilon_{ATE}$ | $\sqrt{\epsilon_{PEHE}}$ | $\epsilon_{ATE}$ | $\sqrt{\epsilon_{PEHE}}$ | $\epsilon_{ATE}$ |
| OLS | $1.13 \pm 0.49$ | $0.30 \pm 0.24$ | $1.14 \pm 0.50$ | $0.30 \pm 0.24$ | $1.17 \pm 0.50$ | $0.30 \pm 0.26$ | $1.36 \pm 0.64$ | $0.35 \pm 0.29$ |
| BART | $4.18 \pm 0.58$ | $0.20 \pm 0.18$ | $4.30 \pm 0.50$ | $0.62 \pm 0.31$ | $4.67 \pm 0.43$ | $1.39 \pm 0.39$ | $5.25 \pm 0.43$ | $2.16 \pm 0.44$ |
| T-learner | $8.32 \pm 0.82$ | $0.40 \pm 0.32$ | $8.88 \pm 0.80$ | $2.77 \pm 0.49$ | $10.18 \pm 0.66$ | $5.44 \pm 0.51$ | $12.14 \pm 0.58$ | $8.16 \pm 0.54$ |
| kNN | $4.82 \pm 0.55$ | $0.41 \pm 0.20$ | $3.96 \pm 0.41$ | $0.20 \pm 0.13$ | $3.94 \pm 0.47$ | $0.65 \pm 0.19$ | $4.49 \pm 0.50$ | $1.15 \pm 0.23$ |
| PSM | $2.05 \pm 0.64$ | $0.53 \pm 0.60$ | $2.76 \pm 0.73$ | $0.61 \pm 0.44$ | $4.25 \pm 0.97$ | $0.92 \pm 0.62$ | $5.91 \pm 1.11$ | $1.99 \pm 0.88$ |
| GANITE | $0.88 \pm 0.08$ | $0.30 \pm 0.23$ | $1.49 \pm 0.03$ | $1.47 \pm 0.06$ | $1.50 \pm 0.01$ | $1.51 \pm 0.01$ | $1.52 \pm 0.02$ | $1.50 \pm 0.01$ |
| DKLite | $0.63 \pm 0.12$ | $0.08 \pm 0.04$ | $2.78 \pm 0.69$ | $2.51 \pm 0.78$ | $5.87 \pm 0.68$ | $5.76 \pm 0.66$ | $8.99 \pm 0.59$ | $8.92 \pm 0.60$ |
| TARNet | $0.55 \pm 0.05$ | $0.54 \pm 0.05$ | $0.55 \pm 0.04$ | $0.53 \pm 0.05$ | $0.54 \pm 0.06$ | $0.52 \pm 0.06$ | $0.55 \pm 0.05$ | $0.54 \pm 0.05$ |
| DragonNet | $\underline{0.29 \pm 0.08}$ | $\underline{0.06 \pm 0.04}$ | $0.51 \pm 0.21$ | $\underline{0.06 \pm 0.04}$ | $\underline{0.39 \pm 0.13}$ | $\mathbf{0.07 \pm 0.05}$ | $0.47 \pm 0.22$ | $\underline{0.11 \pm 0.09}$ |
| BNN | $0.43 \pm 0.00$ | $0.40 \pm 0.00$ | $\underline{0.48 \pm 0.01}$ | $0.46 \pm 0.01$ | $0.44 \pm 0.01$ | $0.41 \pm 0.01$ | $\underline{0.44 \pm 0.01}$ | $0.42 \pm 0.01$ |
| $CFR_{Wass}$ | $0.53 \pm 0.03$ | $0.50 \pm 0.03$ | $0.53 \pm 0.04$ | $0.51 \pm 0.05$ | $0.54 \pm 0.07$ | $0.52 \pm 0.07$ | $0.55 \pm 0.05$ | $0.53 \pm 0.05$ |
| $CFR_{MMD}$ | $0.51 \pm 0.03$ | $0.49 \pm 0.03$ | $0.51 \pm 0.03$ | $0.49 \pm 0.03$ | $0.50 \pm 0.07$ | $0.48 \pm 0.08$ | $1.05 \pm 1.68$ | $0.59 \pm 0.37$ |
| CFR-Pro | $0.50 \pm 0.40$ | $0.49 \pm 0.39$ | $0.51 \pm 0.42$ | $0.50 \pm 0.40$ | $0.57 \pm 0.43$ | $0.55 \pm 0.45$ | $0.73 \pm 0.14$ | $0.46 \pm 0.10$ |
| CE-RCFR | $0.49 \pm 0.01$ | $0.45 \pm 0.01$ | $0.52 \pm 0.01$ | $0.50 \pm 0.01$ | $0.56 \pm 0.43$ | $0.55 \pm 0.45$ | $0.51 \pm 0.02$ | $0.49 \pm 0.02$ |
| DESCN | $0.51 \pm 0.01$ | $0.49 \pm 0.04$ | $0.52 \pm 0.05$ | $0.49 \pm 0.01$ | $0.52 \pm 0.06$ | $0.49 \pm 0.05$ | $0.51 \pm 0.06$ | $0.50 \pm 0.01$ |
| ESCFR | $0.63 \pm 0.16$ | $0.10 \pm 0.07$ | $0.78 \pm 0.20$ | $0.24 \pm 0.09$ | $0.99 \pm 0.25$ | $0.41 \pm 0.12$ | $0.99 \pm 0.25$ | $0.41 \pm 0.12$ |
| G-learner | $\mathbf{0.21 \pm 0.03}$ | $\mathbf{0.05 \pm 0.04}$ | $\mathbf{0.23 \pm 0.05}$ | $\mathbf{0.05 \pm 0.05}$ | $\mathbf{0.30 \pm 0.07}$ | $\underline{0.09 \pm 0.05}$ | $\mathbf{0.36 \pm 0.15}$ | $\mathbf{0.10 \pm 0.11}$ |

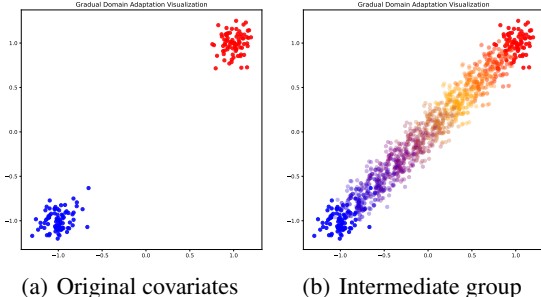

(a) Original covariates      (b) Intermediate group

Figure 2: Illustration results of the original covariates (left) and generated intermediate groups (right). Blue nodes in the lower-left are control samples, red nodes in the upper-right are treated samples.

the performance of our method improves in terms of both PEHE and ATE, which indicates that the prediction models can be smoothly adjusted from one group to another, enhancing the performance of causal inference. As the number of intermediate groups increases, much more samples are generated, which could amplify the influence of label noise, thereby resulting in a slight performance degradation. Nevertheless, the results remain relatively stable across different values of $K$. Figure 3(c) and Figure 3(d) present the results with different filter ratios $r$ for the intermediate data. In general, our method consistently achieves promising performance in terms of both evaluation metrics with respect to varying values of $r$.

In addition to simulation data, we also report the results of $\epsilon_{PEHE}$ on News dataset under different parameter in Figure 4. First, in figure 4(a), we observe that when $K = 0$, the model fails to achieve satisfactory performance. In contrast, the model attains its best performance when $K = 2$ and $K = 4$ on $\epsilon_{PEHE}$ in in-sample setting and $\epsilon_{PEHE}$ in out-sample setting, as the discrepancy between the two groups is not excessively large. Therefore, a relatively small value of $K$ is sufficient to yield good results. As $K$ continues to increase, the performance tends to stabilize; however, an overly large $K$ may amplify the influence of noise, leading to a slight degradation in the final performance. Overall, the results remain relatively stable across different values of $K$. Second, in figure 4(b), it can be observed that both $\epsilon_{PEHE}^{in}$ and $\epsilon_{PEHE}^{out}$ achieve their best performance when $r = 0.6$. As $r$ gradually decreases, the number of samples in the intermediate group is reduced, and the model may even be trained only on real samples, resulting in suboptimal performance. On the other hand, if $r$ is too large, an excessive number of low-confidence samples will be introduced, which can also impair the model's predictive ability and thus degrade the performance.

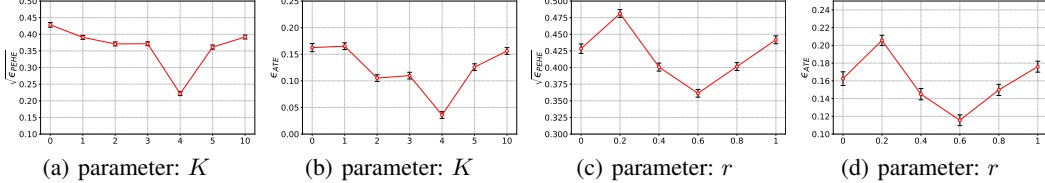

Figure 3: Results including error bar of varying values of the hyperparameters on simulation data ($m_c = 1.4$).

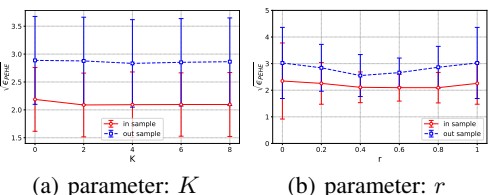

Figure 4: Results including error bar of varying values of the hyperparameters on News data.

### B.7 ABLATION STUDY

The ablation study results about G-learner without OT geodesic (w/o OT) or without data filtering (w/o filtering) are given as Table 5. The results demonstrate the effectiveness of the modules.

### B.8 OPTIMAL TRANSPORT WITH ESTIMATED MARGINAL DISTRIBUTIONS

In this experiment, we modify the marginal distribution from uniform distributions to estimated distributions based on kernel density estimation, and report the results in Table 6 and 7. Overall, the two versions have similar performance. Due to the estimation difficulty of the marginal distribution, the version with uniform distributions performs slightly better.

### B.9 COMPUTATIONAL COST

Let $n_0$, $n_1$, and $d$ be the numbers of control samples, treated samples, and features, respectively. According to Cuturi (2013), the complexity of optimal transport in Eq. (8) is in $O((n_0 + n_1)n_0 n_1 log(n_0 + n_1) + n_0 n_1 d)$. Fortunately, we only need to solve one optimal transport problem regardless of the number of intermediate groups. In addition, optimal transport can be approximately solved by the Sinkhorn algorithm (Cuturi, 2013), whose complexity is in $O(T n_0 n_1 + n_0 n_1 d)$, where $T$ is the number of iterations. For each intermediate group, the complexity of calculating the barycenter in Eq. (9) is in $O((n_0 + n_1)d)$. We also compare the running time with other representation-based methods on simulation data in Table 8. Our method exhibits comparable runtime to classical representation-based approaches, and outperforms certain baselines by a slight margin.

### B.10 MORE DATASETS

We also compare our method with IHDP and ACIC in Table 9. The two datasets are widely used in causal inference (Brooks-Gunn et al., 1992; Dorie et al., 2019). Our method achieves highly competitive performance on these datasets.

### B.11 VISUALIZATION OF PREDICTION MODEL ADJUSTMENT

We also visualize the adjustment procedure of the outcome prediction model from the control group to the treated group in Figure 5. We generate synthetic data on a one-dimensional (1D) dataset,

Table 5: Out-sample performance comparison of ablation methods on News, Twins, and Jobs datasets (mean ± standard deviation). Lower is better.

| Method | News ($\epsilon_{PEHE}$) | News ($\epsilon_{ATE}$) | Twins ($\epsilon_{PEHE}$) | Twins ($\epsilon_{ATE}$) | Jobs ($R_{POL}$) | Jobs ($\epsilon_{ATT}$) |
|---|---|---|---|---|---|---|
| w/o OT | $2.85 \pm 0.83$ | $0.28 \pm 0.25$ | $0.32 \pm 0.01$ | $0.02 \pm 0.01$ | $0.26 \pm 0.05$ | $0.09 \pm 0.08$ |
| w/o filtering | $2.89 \pm 0.83$ | $0.29 \pm 0.24$ | $0.32 \pm 0.01$ | $0.01 \pm 0.01$ | $0.19 \pm 0.07$ | $0.19 \pm 0.26$ |
| G-learner | $2.87 \pm 0.87$ | $0.25 \pm 0.20$ | $0.32 \pm 0.01$ | $0.01 \pm 0.01$ | $0.17 \pm 0.06$ | $0.06 \pm 0.07$ |

Table 6: Out-sample performance comparison of OT with KDE on News, Twins, and Jobs datasets (mean ± standard deviation). Lower is better.

| Method | News ($\epsilon_{PEHE}$) | News ($\epsilon_{ATE}$) | Twins ($\epsilon_{PEHE}$) | Twins ($\epsilon_{ATE}$) | Jobs ($R_{POL}$) | Jobs ($\epsilon_{ATT}$) |
|---|---|---|---|---|---|---|
| OT with KDE | $2.87 \pm 0.78$ | $0.24 \pm 0.30$ | $0.32 \pm 0.01$ | $0.02 \pm 0.02$ | $0.17 \pm 0.10$ | $0.23 \pm 0.05$ |
| G-learner | $2.87 \pm 0.87$ | $0.25 \pm 0.20$ | $0.32 \pm 0.01$ | $0.01 \pm 0.01$ | $0.17 \pm 0.06$ | $0.06 \pm 0.07$ |

where the means of the control group and the treated group are set to 0 and 0.5, respectively. The outcome is generated as $Y_0 = -x^2 + W_1 x + 3 + \varepsilon$ where $W_1 \sim \mathcal{N}(1, 0.1)$ and $\varepsilon \sim \mathcal{N}(0, 0.1)$). We observe that the outcomes are transported from the control group to the treated group smoothly.

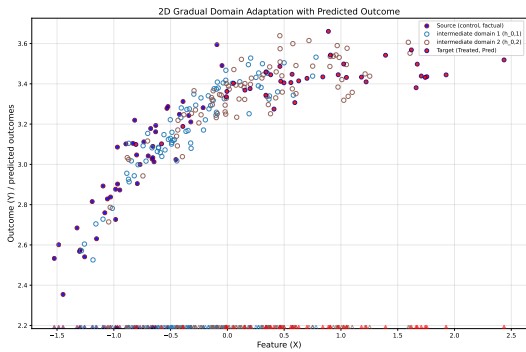

Figure 5: Visualization of the outcome model adjustment.

## C  Proof of Lemma 1

**Lemma 1** *Following (He et al., 2024), consider two arbitrary measures from group $\kappa$ and $\kappa^-$, the error of $|\mathcal{E}(h_{0,\kappa}) - \mathcal{E}(h_{0,\kappa^-})|$ is upper bounded with probability at least $1 - r$ as:*

$$|\mathcal{E}(h_{0,\kappa}) - \mathcal{E}(h_{0,\kappa^-})| \leq \mathcal{O}(\mathcal{W}_p(\mu_\kappa, \mu_{\kappa^-}) + \frac{\rho C + \sqrt{\log(1/r)}}{\sqrt{n}}). \quad (17)$$

Firstly, we have the following assumption:

**Assumption 4** (R-Lipschitz). We assume $\mathcal{H}$ be a class of regressors $h : \mathcal{X} \to [a, b]$ is R-Lipschitz in $\ell_2$ norm, *i.e.*, $\forall h \in \mathcal{H}, \forall x, x' \in \mathcal{X}$:

$$|h(x) - h(x')| \leq R\|x - x'\|_2. \quad (18)$$

**Assumption 5** ($\rho$-Lipschitz Loss). We assumption the loss function $\ell$ is $\rho$-Lipschitz in each argument, i.e. $\forall y, y' \in \mathcal{Y}$:

$$|\ell(y, \cdot) - \ell(y', \cdot)| \leq \rho|y - y'|, \quad (19)$$

$$|\ell(\cdot, y) - \ell(\cdot, y')| \leq \rho|y - y'|. \quad (20)$$

Table 7: In-sample performance comparison of OT with KDE on News, Twins, and Jobs datasets (mean ± standard deviation). Lower is better.

| Method | News ($\epsilon_{PEHE}$) | News ($\epsilon_{ATE}$) | Twins ($\epsilon_{PEHE}$) | Twins ($\epsilon_{ATE}$) | Jobs ($R_{POL}$) | Jobs ($\epsilon_{ATT}$) |
|---|---|---|---|---|---|---|
| OT with KDE | $2.29 \pm 0.57$ | $0.35 \pm 0.26$ | $0.32 \pm 0.00$ | $0.02 \pm 0.02$ | $0.21 \pm 0.06$ | $0.17 \pm 0.08$ |
| G-learner | $2.28 \pm 0.74$ | $0.23 \pm 0.15$ | $0.32 \pm 0.00$ | $0.01 \pm 0.00$ | $0.21 \pm 0.02$ | $0.06 \pm 0.03$ |

Table 8: Running time comparison of different methods.

| Method | Running Time (s) |
|---|---|
| $CFR_{wass}$ | 280.447 |
| $CFR_{mmd}$ | 250.137 |
| EsCFR | 261.413 |
| GANITE | 38.8825 |
| DKLite | 20.369 |
| DragonNet | 254.232 |
| BNN | 32.709 |
| G-learner | 227.049 |

Table 9: Results comparison on the IHDP and ACIC datasets (mean ± standard deviation). The best results are in bold, and the second-best results are underlined.

| Model | IHDP | | | | ACIC | | | |
|---|---|---|---|---|---|---|---|---|
| | $\epsilon_{PEHE}^{in}$ | $\epsilon_{PEHE}^{out}$ | $\epsilon_{ATE}^{in}$ | $\epsilon_{ATE}^{out}$ | $\epsilon_{PEHE}^{in}$ | $\epsilon_{PEHE}^{out}$ | $\epsilon_{ATE}^{in}$ | $\epsilon_{ATE}^{out}$ |
| EsCFR | **1.1676 ± 1.1347** | 1.2581 ± 1.3166 | 0.2564 ± 0.3886 | 0.4048 ± 0.5693 | 0.1639 ± 0.0172 | 0.1654 ± 0.0203 | 0.0420 ± 0.0079 | 0.0374 ± 0.0076 |
| $CFR_{Wass}$ | 1.2879 ± 0.2508 | 1.3431 ± 0.2154 | 0.2158 ± 0.1583 | **0.3002 ± 0.1883** | 0.5035 ± 0.0154 | 0.5063 ± 0.0155 | 0.0370 ± 0.0080 | 0.0361 ± 0.0063 |
| $CFR_{MMD}$ | 1.2075 ± 0.9271 | 1.3160 ± 1.0927 | **0.2104 ± 0.1267** | 0.3749 ± 0.2941 | 0.5034 ± 0.0138 | 0.5062 ± 0.0137 | **0.0121 ± 0.0077** | **0.0108 ± 0.0066** |
| DragonNet | 3.4911 ± 5.2079 | 3.2626 ± 4.8384 | 1.3069 ± 2.2908 | 1.9523 ± 3.5459 | 0.4709 ± 0.1196 | 0.4419 ± 0.1192 | 0.1465 ± 0.1042 | 0.1410 ± 0.0859 |
| GANITE | 8.2262 ± 7.4578 | 5.5415 ± 1.2962 | 7.7728 ± 6.1049 | 6.4089 ± 3.7900 | 0.3276 ± 0.0287 | 0.2995 ± 0.0226 | 0.2366 ± 0.0129 | 0.2364 ± 0.0129 |
| DKLite | 4.4072 ± 7.4140 | 3.7288 ± 5.7008 | 0.2953 ± 0.2497 | 0.9048 ± 1.8252 | 0.1809 ± 0.0086 | 0.1996 ± 0.0084 | 0.0331 ± 0.0112 | 0.0299 ± 0.0110 |
| G-learner | 1.1888 ± 0.7051 | **1.1952 ± 1.6739** | 0.3108 ± 0.2978 | 0.3738 ± 0.2959 | **0.1569 ± 0.0106** | **0.1599 ± 0.0117** | 0.0287 ± 0.0087 | 0.0298 ± 0.0107 |

**Assumption 6** (Bounded Model Complexity). We assume the Rademacher complexity (Bartlett & Mendelson, 2002), $\mathcal{R}$, of the hypothesis class, $\mathcal{H}$, is bounded for any distribution $\mu$ considered in this paper. That is, for some constant $C \geq 0$,

$$\mathcal{R}_n(\mathcal{H}; \mu) = \mathbb{E}\left[\sup_{h \in \mathcal{H}} \frac{1}{n} \sum_{i=1}^{n} \sigma_i h(\mathbf{x}_i)\right] \leq \frac{C}{\sqrt{n}}, \tag{21}$$

where the expectation is w.r.t. $\mathbf{x}_i \sim \mu_{\mathcal{X}}$ and $\sigma_i \sim \text{Uniform}(\{-1, 1\})$ for $i = 1, \ldots, n$.

*Proof.* For any predictor $h$, the risk difference is

$$|\mathcal{E}_\kappa(h) - \mathcal{E}_{\kappa^-}(h)| = |\int_{\mathbf{x} \in \{\mathbf{x}_\kappa\}} (h(\mathbf{x}) - y)^2 p(\mathbf{x}) d\mathbf{x} - \int_{\mathbf{x}' \in \{\mathbf{x}_{\kappa^-}\}} (h(\mathbf{x}') - y')^2 p(\mathbf{x}) d\mathbf{x}|. \tag{22}$$

Introduce a coupling $\gamma$ of $\mu_\kappa$ and $\mu_{\kappa^-}$.

$$|\mathcal{E}_\kappa(h) - \mathcal{E}_{\kappa^-}(h)| \leq \int |\ell(h(x), y) - \ell(h(x'), y')| d\gamma$$
$$\leq \rho(R\|\mathbf{x} - \mathbf{x}'\| + \|y - y'\|) \quad \text{(by Assumption 4 and Assumption 5)}$$
$$\leq \rho R \mathcal{W}_p(\mu_\kappa, \mu_{\kappa^-}). \tag{23}$$

So that we can get,

$$|\mathcal{E}(h_{0,\kappa}) - \mathcal{E}(h_{0,\kappa^-})| \leq \mathcal{O}(\mathcal{W}_p(\mu_\kappa, \mu_{\kappa^-}) + \frac{\rho C + \sqrt{\log(1/r)}}{\sqrt{n}}). \tag{24}$$

Specific details of the proof above is similar to the proof of proposition 1 in (He et al., 2024).

$\square$

## D    PROOF OF LEMMA 2

**Lemma 2** *For a filtering ratio $r \in (0, 1)$, the loss $\mathcal{E}(h_{0,1})$ is upper bounded with probability at least $1 - r$ as*

$$\mathcal{E}(h_{0,1}) \leq \mathcal{E}(h_{0,0}) + \mathcal{O}(\tilde{K}\Delta + \frac{\tilde{K}}{\sqrt{n}} + \tilde{K}\sqrt{\frac{\log(1/r)}{n}} + \frac{1}{\sqrt{n\tilde{K}}} + \sqrt{\frac{(\log n\tilde{K})^{3L-2}}{n\tilde{K}}} + \sqrt{\frac{\log(1/r)}{n\tilde{K}}}),$$

*where $n$ is the number of training samples, $L$ is the model depth, and $\tilde{K}\Delta$ is the accumulated distribution shift between the control and treated groups through intermediate groups*

*Proof.* Treat gradual self-training over $\tilde{K} + 1$ domains (each with $n$ points) as running an online learner for $n\tilde{K}$ sequential examples. Directly invoke Corollary 2 of (Kuznetsov & Mohri, 2020) to get, with probability $1 - r$, following (He et al., 2024),

$$\mathcal{E}(h_{0,1}) \leq \sum_{k=0}^{\tilde{K}} \sum_{i=0}^{n} q_{kn+i} \mathcal{E}(h_{0,\frac{k}{\tilde{K}}}) + \text{disc}(\mathbf{q}_{n(\tilde{K}+1)})$$

$$+ \|\mathbf{q}_{n(\tilde{K}+1)}\|_2 + 6C\sqrt{4\pi \log(n\tilde{K})} R_{nK}^{\text{seq}}(\ell \circ \mathcal{H}) + C \|q_{n(\tilde{K}+1)}\|_2 \sqrt{\frac{8 \ln(1/r)}{n\tilde{K}}}.$$

(25)

where we index each group by $\kappa = \frac{k}{K+1}$, where $k = 0, 1, \ldots, K + 1$. $\mathcal{H}$ denotes the hypothesis class (the set of all predictors your algorithm may choose from), and $C$ is the uniform upper bound on the loss function $\ell$, i.e. $\ell(h(x), y) \in [0, C]$ for all $h \in \mathcal{H}$ and all $(x, y)$, $\text{disc}(\cdot)$ denotes the discrepancy term as defined in (Kuznetsov & Mohri, 2020), and $q_{n(\tilde{K}+1)}$ is taken as $\mathbf{q}_{n(\tilde{K}+1)} = q_{n(\tilde{K}+1)}^* = \left( \frac{1}{n(\tilde{K}+1)}, \ldots, \frac{1}{n(\tilde{K}+1)} \right)$. From Lemma 1 from (He et al., 2024) and Lemma 1 we know:

$$\sum_{k=0}^{\tilde{K}} \sum_{i=0}^{n} q_{kn+i} \mathcal{E}(h_{0,\frac{k}{\tilde{K}}}) \leq \mathcal{E}(h_{0,0}) + \mathcal{O}(\tilde{K}\Delta) + \mathcal{O}(\frac{\tilde{K}}{\sqrt{n}} + \tilde{K}\sqrt{\frac{\log\frac{1}{r}}{n}}).$$

(26)

From Lemma 2 from (He et al., 2024) and Lemma 1 we know $\text{disc}(q_{n(\tilde{K}+1)})$ can be bound. From Lemma 4 and Example 2 on neural-network sequence complexity from (He et al., 2024) we know:

$$6C \sqrt{4\pi \log(n\tilde{K})} R_{n\tilde{K}}^{\text{seq}}(\ell \circ \mathcal{H}) \leq \mathcal{O}\Big( \sqrt{\frac{(\log n\tilde{K})^{3L-2}}{n\tilde{K}}} \Big).$$

(27)

In conclusion,

$$\mathcal{E}(h_{0,1}) \leq \mathcal{E}(h_{0,0}) + \mathcal{O}(\tilde{K}\Delta + \frac{\tilde{K}}{\sqrt{n}} + \tilde{K}\sqrt{\frac{\log(1/r)}{n}} + \frac{1}{\sqrt{n\tilde{K}}} + \sqrt{\frac{(\log n\tilde{K})^{3L-2}}{n\tilde{K}}} + \sqrt{\frac{\log(1/r)}{n\tilde{K}}}),$$

$\square$

## E    PROOF OF THEOREM 1

**Theorem 1** *The effect estimation error $\epsilon_{PEHE}$ is upper bounded by:*

$$\epsilon_{PEHE} \leq 2\mathcal{E}(h_{0,0}) + 2\mathcal{E}(h_{1,1}) + 2\mathcal{O}(\tilde{K}\Delta + \mathcal{B}(n, \tilde{K}, L, r)),$$

*where*

$$\mathcal{B}(n, \tilde{K}, L, r) = \frac{\tilde{K}}{\sqrt{n}} + \tilde{K}\sqrt{\frac{\log(1/r)}{n}} + \frac{1}{\sqrt{n\tilde{K}}} + \sqrt{\frac{(\log n\tilde{K})^{3L-2}}{n\tilde{K}}} + \sqrt{\frac{\log(1/r)}{n\tilde{K}}}.$$

*Proof.*

$$\epsilon_{PEHE} = \int_{\mathcal{X}} (\hat{\tau}(\mathbf{x}) - \tau(\mathbf{x}))^2 p(\mathbf{x})d\mathbf{x} = \int_{\mathcal{X}} ((h_1(\mathbf{x}) - h_0(\mathbf{x})) - (f_1(\mathbf{x}) - f_0(\mathbf{x})))^2 p(\mathbf{x})d\mathbf{x}$$

$$= \int_{\mathcal{X}} ((h_1(\mathbf{x}) - f_1(\mathbf{x})) + (f_0(\mathbf{x}) - h_0(\mathbf{x})))^2 p(\mathbf{x})d\mathbf{x}$$

$$\leq \int_{\mathcal{X}} ((h_1(\mathbf{x}) - f_1(\mathbf{x}))^2 + (f_0(\mathbf{x}) - h_0(\mathbf{x}))^2) p(\mathbf{x})d\mathbf{x}$$

$$= \int_{\mathcal{X}} ((h_1(\mathbf{x}) - f_1(\mathbf{x}))^2 p(\mathbf{x}|t = 1)d\mathbf{x}$$

$$+ \int_{\mathcal{X}} ((h_1(\mathbf{x}) - f_1(\mathbf{x}))^2 p(\mathbf{x}|t = 0)d\mathbf{x}$$

$$+ \int_{\mathcal{X}} ((h_0(\mathbf{x}) - f_0(\mathbf{x}))^2 p(\mathbf{x}|t = 1)d\mathbf{x}$$

$$+ \int_{\mathcal{X}} ((h_0(\mathbf{x}) - f_0(\mathbf{x}))^2 p(\mathbf{x}|t = 0)d\mathbf{x}$$

$$= \mathcal{E}(h_{1,1}) + \mathcal{E}(h_{1,0}) + \mathcal{E}(h_{0,1}) + \mathcal{E}(h_{0,0}). \tag{28}$$

**Lemma 2** *For a filtering ratio $r \in (0, 1)$, the loss $\mathcal{E}(h_{0,1})$ is upper bounded with probability at least $1 - r$ as*

$$\mathcal{E}(h_{0,1}) \leq \mathcal{E}(h_{0,0}) + \mathcal{O}(\tilde{K}\Delta + \frac{\tilde{K}}{\sqrt{n}} + \tilde{K}\sqrt{\frac{\log(1/r)}{n}} + \frac{1}{\sqrt{n\tilde{K}}} + \sqrt{\frac{(\log n\tilde{K})^{3L-2}}{n\tilde{K}}} + \sqrt{\frac{\log(1/r)}{n\tilde{K}}}), \tag{29}$$

*where $n$ is the number of training samples, $L$ is the model depth, and $\tilde{K}\Delta$ is the accumulated distribution shift between the control and treated groups through intermediate groups*
According to Lemma 1, it is easy to get:

$$\mathcal{E}(h_{0,1}) \leq \mathcal{E}(h_{0,0}) + \mathcal{O}(\tilde{K}\Delta + \frac{\tilde{K}}{\sqrt{n}} + \tilde{K}\sqrt{\frac{\log(1/r)}{n}} + \frac{1}{\sqrt{n\tilde{K}}} + \sqrt{\frac{(\log n\tilde{K})^{3L-2}}{n\tilde{K}}} + \sqrt{\frac{\log(1/r)}{n\tilde{K}}}),$$

$$\mathcal{E}(h_{1,0}) \leq \mathcal{E}(h_{1,1}) + \mathcal{O}(\tilde{K}\Delta + \frac{\tilde{K}}{\sqrt{n}} + \tilde{K}\sqrt{\frac{\log(1/r)}{n}} + \frac{1}{\sqrt{n\tilde{K}}} + \sqrt{\frac{(\log n\tilde{K})^{3L-2}}{n\tilde{K}}} + \sqrt{\frac{\log(1/r)}{n\tilde{K}}}),$$

Then, combine Eq.(28), we can obtain the upper bound of $\epsilon_{PEHE}$:

$$\epsilon_{PEHE} \leq \mathcal{E}(h_{1,1}) + \mathcal{E}(h_{1,0}) + \mathcal{E}(h_{0,1}) + \mathcal{E}(h_{0,0})$$

$$\leq 2\mathcal{E}(h_{1,1}) + 2\mathcal{E}(h_{0,0})$$

$$+ 2\mathcal{O}(\tilde{K}\Delta + \frac{\tilde{K}}{\sqrt{n}} + \tilde{K}\sqrt{\frac{\log(1/r)}{n}} + \frac{1}{\sqrt{n\tilde{K}}} + \sqrt{\frac{(\log n\tilde{K})^{3L-2}}{n\tilde{K}}} + \sqrt{\frac{\log(1/r)}{n\tilde{K}}}). \tag{30}$$

$\square$

