# OpenReview forum: "Adjusting Prediction Model Through Wasserstein Geodesic for Causal Inference"
_ICLR.cc/2026/Conference — ICLR 2026 Poster_

### Official Review · Reviewer_JEDq · 2025-10-28

**Soundness:** 3
**Presentation:** 3
**Contribution:** 3
**Rating:** 6
**Confidence:** 2

**Summary:**

The paper addresses covariate shift between treated and control groups in observational causal inference. Instead of balancing covariates via reweighting/representation learning (which can cause over-balancing and remove predictive information), the authors propose to adjust the outcome prediction models themselves. They:

- Build an optimal transport plan between the empirical control and treated distributions and use the Wasserstein geodesic (displacement interpolation) to generate K intermediate “groups” µκ that smoothly connect the two.
- Perform gradual self-training along this path: starting from a supervised model trained on one endpoint (e.g., h0,0 on control), iteratively pseudo-label the next intermediate group and fit a new model h0,κ, until reaching the other endpoint (yielding h0,1). A symmetric process is used for h1.
- Use MC-Dropout uncertainty to filter the generated samples (keep the r fraction with lowest predictive std) to mitigate pseudo-label noise.
- Provide generalization bounds that adapt gradual domain adaptation theory (He et al., 2024) to their regression setting, bounding E(h0,1) and PEHE in terms of endpoint errors, the average inter-step Wasserstein distance Δ and finite-sample/model-complexity terms.
- Empirically evaluate on News, Twins, Jobs, and a synthetic suite with increasing group shift, showing competitive or state-of-the-art performance on most metrics, especially as confounding grows.

**Strengths:**

- **Originality and framing**
  Conceptual shift from "adjust covariates" to "adjust the prediction model" via a smooth geodesic path is a clear and appealing alternative to conventional balancing. This directly targets the over-balancing concern. Using OT-induced displacement interpolation to construct intermediate "domains" is principled and leverages known geometry of probability measures.

- **Methodological quality**
  The algorithmic pipeline is simple, modular, and practical: compute OT; generate intermediate groups; gradual self-training with uncertainty filtering. It can wrap around standard regressors. The uncertainty-based filtering is a reasonable, low-overhead safeguard against error amplification in self-training.

- **Theoretical and Empirical Support**
  The claims are well-supported by both theoretical analysis and extensive empirical evidence. The authors provide a theoretical upper bound on the treatment effect estimation error (Theorem 1), which solidifies the method's foundations. The experimental evaluation is comprehensive, covering multiple datasets, evaluation metrics, and a large number of strong baselines. The consistently superior performance of G-learner across different scenarios robustly demonstrates its effectiveness.

**Weaknesses:**

- **Scope and assumptions**
  The approach still relies on ignorability/overlap and does not address unobserved confounding. While this is standard for meta-learners, the paper’s emphasis on avoiding balancing could be read as implying robustness to confounding; a clearer articulation of identifiability limits would help.

- **Discrete vs. Continuous Covariates**
  The use of Wasserstein distance with an L2 cost function seems most natural for continuous covariates. It is not immediately clear how the method would perform with datasets dominated by discrete or categorical features. While the method appears to work well on the presented real-world datasets which likely contain mixed data types, a brief discussion on the applicability or potential adaptations for high-dimensional discrete covariate spaces would be beneficial.

- **Experimental gaps**
  - On News, G-learner's PEHE is worse than TARNet; the paper states best or competitive, but this discrepancy merits discussion and an ablation (e.g., how K, r, or representation choice affects PEHE on News).
  - Dataset breadth: adding IHDP/ACIC or a modern semi-synthetic benchmark would strengthen generality claims.
  - Computational cost: solving the OT problem can be heavy for large n.

**Questions:**

- **Necessity of OT geodesics:**
  How much of the gain comes from using the OT coupling versus any random or nearest-neighbor pairing for interpolation? Would performing OT in a learned latent space (e.g., via an autoencoder or TARNet trunk) further improve the realism of intermediate groups? Any preliminary results?

- **Computational Cost:**
  Could the authors please comment on the computational complexity of the proposed G-learner, particularly the optimal transport step?

- **Visualization of Model Adjustment:**
  The visualization of the generated intermediate data in Figure 2 is very effective. It would be highly insightful to also visualize the function learned by the outcome models (e.g., h_0,k) at each step k. For a simple 1D or 2D covariate space, plotting how the prediction surface smoothly deforms from the control group towards the treated group would provide a powerful visual confirmation of the "gradual model adjustment" concept. Is this something the authors have considered or could provide?

---

> ### Author Response · Authors · 2025-11-27
>
> We sincerely appreciate the insightful feedback from the reviewers and present our detailed responses below. The manuscript will be revised accordingly based on both the reviewers’ comments and our replies.
>
> **Weakness 1:**
> Discussion on assumptions and identifiability.
>
> **Answer:**
> We thank the reviewer for pointing out this.
> Our G-learner indeed does not relax traditional assumptions, including ignorability and overlap.
> Instead, within the standard identification framework, we focus on avoiding the over-balancing issue in existing balanced representation learning methods through a model adjustment method.
> We will clarify this in the revised version.
>
> **Weakness 2:**
> Discussion on the applicability of discrete covariate spaces.
>
> **Answer:**
> Thank you for the valuable comment.
> Indeed, optimal transport requires a well-defined cost function, thus is suitable for continuous covariates.
> For discrete or categorical data, there are two potential approaches:
> (1) One can define a distance for discrete or categorical data, so that optimal transport can be applied on the defined distance.
> (2) It is possible to learn a mapping function to project discrete or categorical data into a representation space, and then perform optimal transport.
> We will discuss this in the revision.
>
> **Weakness 3.1:**
> Parameter sensitivity of PEHE on News dataset.
>
> **Answer:**
> Thank you for the insightful comment.
> We show the result of PEHE in different parameter on News datasets as follows.
>
> First, it can be observed that when $K = 0$,
> the model fails to achieve satisfactory performance.
> In contrast, the model attains its best performance when $K = 2$ and $K = 4$ on PEHE\_in and PEHE\_out,
> as the discrepancy between the two groups is not excessively large.
> Therefore, a relatively small value of $K$ is sufficient to yield good results.
> As $K$ continues to increase,
> the performance tends to stabilize;
> however, an overly large $K$ may amplify the influence of noise,
> leading to a slight degradation in the final performance.
> Overall, the results remain relatively stable across different values of $K$.
>
> |          | K=0             | K=2             | K=4             | K=6             | K=8             |
> |----------|-----------------|-----------------|-----------------|-----------------|-----------------|
> | PEHE\_in  | 2.1875 ± 0.5718 | 2.0878 ± 0.5713 | 2.0929 ± 0.5866 | 2.0958 ± 0.5678 | 2.0961 ± 0.5731 |
> | PEHE\_out | 2.8871 ± 0.7880 | 2.8773 ± 0.7839 | 2.8333 ± 0.7844 | 2.8526 ± 0.7838 | 2.8636 ± 0.7847 |
>
>
> Second, it can be observed that both PEHE\_in and PEHE\_out achieve their best performance when $r = 0.6$.
> As $r$ gradually decreases,
> the number of samples in the intermediate group is reduced,
> and the model may even be trained only on real samples.
> This can lead to underfitting, resulting in suboptimal performance.
> On the other hand, if $r$ is too large, an excessive number of low-confidence samples will be introduced,
> which can also impair the model’s predictive ability and thus degrade the performance.
>
> |          | r=0             | r=0.2           | r=0.4           | r=0.6           | r=0.8           | r=1             |
> |----------|-----------------|-----------------|-----------------|-----------------|-----------------|-----------------|
> | PEHE\_in  | 2.3465 ± 1.4294 | 2.2569 ± 0.7832 | 2.1097 ± 0.5771 | 2.0957 ± 0.5048 | 2.0961 ± 0.5731 | 2.2569 ± 0.7832 |
> | PEHE\_out | 3.0245 ± 1.3364 | 2.8422 ± 0.8809 | 2.5514 ± 0.7889 | 2.6613 ± 0.5478 | 2.8636 ± 0.7847 | 3.0245 ± 1.3364 |

---

> ### Author Response · Authors · 2025-11-27
>
> **Weakness 3.2:**
> Results on IHDP/ACIC datasets.
>
> **Answer:**
> Thank you for your helpful advice.
> We have added experiments on the IHDP and ACIC dataset as follows.
> Our method achieves highly competitive performance on these datasets.
>
> | IHDP   | pehe_in         | pehe_out        | ate_in          | ate_out         |
> | --------- | --------------- | --------------- | --------------- | --------------- |
> | EsCFR     | 1.1676 ± 1.1347 | 1.2581 ± 1.3166 | 0.2564 ± 0.3886 | 0.4048 ± 0.5693 |
> | CFR_wass  | 1.2879 ± 0.2508 | 1.3431 ± 0.2154 | 0.2158 ± 0.1583 | 0.3002 ± 0.1883 |
> | CFR_MMD   | 1.2075 ± 0.9271 | 1.3160 ± 1.0927 | 0.2104 ± 0.1267 | 0.3749 ± 0.2941 |
> | DragonNet | 3.4911 ± 5.2079 | 3.2626 ± 4.8384 | 1.3069 ± 2.2908 | 1.9523 ± 3.5459 |
> | GANITE    | 8.2262 ± 7.4578 | 5.5415 ± 1.2962 | 7.7728 ± 6.1049 | 6.4089 ± 3.7900 |
> | DKLite    | 4.4072 ± 7.4140 | 3.7288 ± 5.7008 | 0.2953 ± 0.2497 | 0.9048 ± 1.8252 |
> | G-learner | 1.1888 ± 0.7051 | 1.1952 ± 1.6739 | 0.3108 ± 0.2978 | 0.3738 ± 0.2959 |
>
> | ACIC   | pehe_in         | pehe_out        | ate_in          | ate_out         |
> | --------- | --------------- | --------------- | --------------- | --------------- |
> | EsCFR     | 0.1639 ± 0.0172 | 0.1654 ± 0.0203 | 0.0420 ± 0.0079 | 0.0374 ± 0.0076 |
> | CFR_wass  | 0.5035 ± 0.0154 | 0.5063 ± 0.0155 | 0.0370 ± 0.0080 | 0.0361 ± 0.0063 |
> | CFR_MMD   | 0.5034 ± 0.0138 | 0.5062 ± 0.0137 | 0.0121 ± 0.0077 | 0.0108 ± 0.0066 |
> | DragonNet | 0.4709 ± 0.1196 | 0.4419 ± 0.1192 | 0.1465 ± 0.1042 | 0.1410 ± 0.0859 |
> | GANITE    | 0.3276 ± 0.0287 | 0.2995 ± 0.0226 | 0.2366 ± 0.0129 | 0.2364 ± 0.0129 |
> | DKLite    | 0.1809 ± 0.0086 | 0.1996 ± 0.0084 | 0.0331 ± 0.0112 | 0.0299 ± 0.0110 |
> | G-learner | 0.1569 ± 0.0106 | 0.1599 ± 0.0117 | 0.0287 ± 0.0087 | 0.0298 ± 0.0107 |
>
>
> **Weakness 3.3 and Question 2:**
> Computational cost.
>
> **Answer:**
> Let $n_0$, $n_1$, and $d$ be the numbers of control samples, treated samples, and features, respectively.
> According to [a], the complexity of optimal transport in Eq. (8) is in $O((n_0+n_1) n_0 n_1 log(n_0 + n_1) + n_0 n_1 d)$.
> Fortunately, we only need to solve one optimal transport problem regardless of the number of intermediate groups.
> In addition, optimal transport can be approximately solved by the Sinkhorn algorithm [a], whose complexity is in $O(T n_0 n_1 + n_0 n_1 d)$, where $T$ is the number of iteration.
>
> For each intermediate group, the complexity of calculating barycenter in Eq. (9) is in $O((n_0 + n_1)d)$.
>
> We also compare the running time with other representation-based methods on simulation data.
> Our method exhibits comparable runtime to classical representation-based approaches, and even outperforms certain baselines by a slight margin.
> |           | running time |
> |-----------|--------------|
> | CFR\_wass  | 280.447s     |
> | CFR\_mmd   | 250.137s     |
> | EsCFR     | 261.413s     |
> | GANITE    | 38.8825s     |
> | DKLite    | 20.369s      |
> | DragonNet | 254.232s     |
> | BNN       | 32.709s      |
> | G-learner | 227.049s     |
>
>
> **Question 1:**
> Effect of OT geodesic.
>
> **Answer:**
> Thank you for the insightful comment.
> The following is the ablation study results about G-learner without OT geodesic.
> The method w/o geodesic does not leverage the optimal transport plan to generate samples.
> Instead, it generates samples by considering the convex combinations of all the pairs $(x_{0,i}, x_{1,j})$.
> It is easy to seem that the performance get worse without OT geodesic.
>
> | methods         | News_in (pehe)  | News_in (ate)   | Twins_in (pehe) | Twins_in (ate)  | Jobs_in (pr)    | Jobs_in (att)   |
> | ------------ | --------------- | --------------- | --------------- | --------------- | --------------- | --------------- |
> | w/o geodesic | 2.5735 ± 0.6322 | 0.2456 ± 0.0379 | 0.3199 ± 0.0022 | 0.0189 ± 0.0032 | 0.2663 ± 0.0296 | 0.0991 ± 0.0225 |
> | G-learner    | 2.2750 ± 0.7415 | 0.2258 ± 0.1472 | 0.3191 ± 0.0020 | 0.0107 ± 0.0039 | 0.2053 ± 0.0151 | 0.0552 ± 0.0306 |
>
> | methods       | News_out (pehe) | News_out (ate)  | Twins_out (pehe) | Twins_out (ate) | Jobs_out (pr)   | Jobs_out (att)  |
> | ------------ | --------------- | --------------- | ---------------- | --------------- | --------------- | --------------- |
> | w/o geodesic | 2.8506 ± 0.8263 | 0.2832 ± 0.2541 | 0.3212 ± 0.0074  | 0.0243 ± 0.0107 | 0.2601 ± 0.0469 | 0.0902 ± 0.0783 |
> | G-learner    | 2.8681 ± 0.8696 | 0.2451 ± 0.1955 | 0.3200 ± 0.0086  | 0.0084 ± 0.0060 | 0.1691 ± 0.0622 | 0.0596 ± 0.0740 |

---

> > ### Author Response · Authors · 2025-11-27
> >
> > **Question 3:**
> > Visualization of Model Adjustment.
> >
> > **Answer:**
> > Thank you for the helpful suggestion.
> > https://anonymous.4open.science/r/iclr-submit-for-G-learner-3236/2d_domain_evolution_visulization.pdf，
> > We also visualize the geodesic of the outcome transport from control group to treated group.
> > We generate synthetic data on a one-dimensional (1D) dataset, where the means of the control group and the treated group are set to 0 and 0.5, respectively. The outcome is generated as $Y_0 = -x^2+W_1x +3+ \varepsilon$ where $W_1\sim \mathcal N(1,0.1)$ and $\varepsilon\sim \mathcal N(0,0.1))$.
> > We observe that the outcome is transported from the control group to the treated group .
> >
> >
> > [a]Sinkhorn distances: Lightspeed computation of optimal transport, NIPS 2013.

---

### Official Review · Reviewer_c11B · 2025-10-28

**Soundness:** 3
**Presentation:** 3
**Contribution:** 3
**Rating:** 4
**Confidence:** 4

**Summary:**

This paper proposes a novel method, `G-learner`, to address the over-balancing issue during alignment of treated and control groups in causal inference. The authors leverage the Wasserstein space and the concept of the Wasserstein barycenter to generate intermediate domains. Models are then progressively fine-tuned on these intermediate domains. Additionally, an uncertainty quantification technique is introduced to filter data during the fine-tuning stage. Theoretical analyses are provided, and extensive experiments are conducted to demonstrate the efficacy of the proposed approach.

**Strengths:**

1. The use of intermediate domains to generate auxiliary samples for alleviating the over-balancing issue is both creative and effective.
2. The research topic—improving causal inference—is highly relevant to the ICLR community.
3. The experimental results are clearly quantified and support the claims of the proposed approach.

**Weaknesses:**

1. To the best of the reviewer’s knowledge, some related works have explored using the Gromov-Wasserstein distance and unbalanced Wasserstein distance to account for local similarity and mini-batch sampling effect during alignment. Including such baselines (e.g., ESCFR-Pro [1] and CE-RCFR [2]) would strengthen the comparison and evaluation.

2. The paper states that the tables underline the second-best results, but these underlined results are not clearly visible.

3. Figure 3 should include uncertainty/error bars to better demonstrate the robustness of the proposed approach.

4. An ablation study evaluating the effectiveness of the data filtering component is recommended.

5. Is Eq. (9) a variant of barycentric projection? Please clarify how it is derived from Eq. (5).

6. In Figure 3, why does performance decrease as the number of intermediate groups increases?

---
References:
[1]. Proximity Matters: Local Proximity Enhanced Balancing for Treatment Effect Estimation
[2]. CE-RCFR: Robust Counterfactual Regression for Consensus-Enabled Treatment Effect Estimation

**Questions:**

Please see the weaknesses

---

> ### Author Response · Authors · 2025-11-27
>
> We sincerely appreciate the insightful feedback from the reviewers and present our detailed responses below. The manuscript will be revised accordingly based on both the reviewers’ comments and our replies.
>
> **Weakness 1:**
> Comparison with new optimal transport-based methods.
>
> **Answer:**
> Thank you for the helpful suggestion.
> We have added suggested new OT-based methods for comparison in the following.
> We also added a recent baseline DESCN [a] for comparison.
> Our proposed method achieves promising performance compared with baselines.
> | methods      | News_in (pehe)  | News_in (ate)   | Twins_in (pehe) | Twins_in (ate)  | Jobs_in (pr)    | Jobs_in (att)   |
> | --------- | --------------- | --------------- | --------------- | --------------- | --------------- | --------------- |
> | CFR-Pro   | 3.3859 ± 1.3643 | 0.5562 ± 0.4728 | 0.3197 ± 0.0023 | 0.0183 ± 0.0156 | 0.3007 ± 0.0216 | 0.0938 ± 0.0184 |
> | CE-RCER   | 3.2342 ± 1.0884 | 1.3131 ± 0.4016 | 0.3204 ± 0.0027 | 0.0244 ± 0.0195 | 0.2886 ± 0.0378 | 0.1132 ± 0.0524 |
> | DESCN     | 4.3748 ± 1.3147 | 2.6976 ± 0.7238 | 0.3174 ± 0.0031 | 0.0149 ± 0.0105 | 0.2811 ± 0.0349 | 0.1620 ± 0.0178 |
> | G-learner | 2.2750 ± 0.7415 | 0.2258 ± 0.1472 | 0.3191 ± 0.0020 | 0.0107 ± 0.0039 | 0.2053 ± 0.0151 | 0.0552 ± 0.0306 |
>
> | methods      | News_out (pehe) | News_out (ate)  | Twins_out (pehe) | Twins_out (ate) | Jobs_out (pr)   | Jobs_out (att)  |
> | --------- | --------------- | --------------- | ---------------- | --------------- | --------------- | --------------- |
> | CFR-Pro   | 3.3837 ± 1.2978 | 0.5963 ± 0.4602 | 0.3206 ± 0.0088  | 0.0161 ± 0.0133 | 0.2935 ± 0.0623 | 0.0937 ± 0.0863 |
> | CE-RCER   | 3.2956 ± 1.0875 | 1.2729 ± 0.4131 | 0.3214 ± 0.0095  | 0.0244 ± 0.0202 | 0.1781 ± 0.0401 | 0.1265 ± 0.1042 |
> | DESCN     | 4.3308 ± 1.2460 | 2.6741 ± 0.6953 | 0.3244 ± 0.0081  | 0.0168 ± 0.0217 | 0.3011 ± 0.0509 | 0.1920 ± 0.0078 |
> | G-learner | 2.8681 ± 0.8696 | 0.2451 ± 0.1955 | 0.3200 ± 0.0086  | 0.0084 ± 0.0060 | 0.1691 ± 0.0622 | 0.0596 ± 0.0740 |
>
>
> **Weakness 2:**
> Underline the second-best results in tables.
>
> **Answer:**
> Thank you for your the helpful comment.
> We are sorry for the omission. We will correct this in the revision.
>
> **Weakness 3:**
> Error bars in Figure 3.
>
> **Answer:**
> Thank you for your insightful comment.
> We provide the error bar of the Figure 3 as follows.
> It shows that our method achieves robust and promising performance.
>
> | K=1           |               | K=2           |               | K=3           |               | K=4           |               | K=5           |              | K=10          |               |
> |---------------|---------------|---------------|---------------|---------------|---------------|---------------|---------------|---------------|--------------|---------------|---------------|
> | pehe          | ate           | pehe          | ate           | pehe          | ate           | pehe          | ate           | pehe          | ate          | pehe          | ate           |
> | 0.3908±0.0059 | 0.1651±0.0063 | 0.3710±0.0060 | 0.1052±0.0064 | 0.3715±0.0059 | 0.1098±0.0063 | 0.2215±0.0059 | 0.0358±0.0063 | 0.3615±0.0059 | 01258±0.0062 | 0.3918±0.0059 | 0.1561±0.0063 |
>
>
> | r=0           |               | r=0.2         |               | r=0.4         |               | r=0.6         |               | r=0.8         |               | r=1           |               |
> |---------------|---------------|---------------|---------------|---------------|---------------|---------------|---------------|---------------|---------------|---------------|---------------|
> | pehe          | ate           | pehe          | ate           | pehe          | ate           | pehe          | ate           | pehe          | ate           | pehe          | ate           |
> | 0.4284±0.0071 | 0.1625±0.0077 | 0.4812±0.0059 | 0.2056±0.0058 | 0.4007±0.0060 | 0.1450±0.0064 | 0.3613±0.0058 | 0.1156±0.0061 | 0.4015±0.0059 | 0.1498±0.0063 | 0.4417±0.0059 | 0.1760±0.0062 |

---

> ### Author Response · Authors · 2025-11-27
>
> **Weakness 4:**
> Ablation study of data filtering.
>
> **Answer:**
> Thank you for the valuable comments.
> We provide the ablation study of data filtering as follows.
> The results show that data filtering exhibits a substantial enhancement in performance.
>
> | methods         | News_in (pehe)  | News_in (ate)   | Twins_in (pehe) | Twins_in (ate)  | Jobs_in (pr)    | Jobs_in (att)   |
> | ------------- | --------------- | --------------- | --------------- | --------------- | --------------- | --------------- |
> | w/o filtering | 2.3738 ± 0.6638 | 0.2339 ± 0.1709 | 0.3197 ± 0.0020 | 0.0177 ± 0.0067 | 0.2218 ± 0.0554 | 0.1625 ± 0.2185 |
> | G-learner     | 2.2750 ± 0.7415 | 0.2258 ± 0.1472 | 0.3191 ± 0.0020 | 0.0107 ± 0.0039 | 0.2053 ± 0.0151 | 0.0552 ± 0.0306 |
>
> | methods         | News_out (pehe) | News_out (ate)  | Twins_out (pehe) | Twins_out (ate) | Jobs_out (pr)   | Jobs_out (att)  |
> | ------------- | --------------- | --------------- | ---------------- | --------------- | --------------- | --------------- |
> | w/o filtering | 2.8906 ± 0.8263 | 0.2948 ± 0.2417 | 0.3208 ± 0.0086  | 0.0140 ± 0.0067 | 0.1871 ± 0.0664 | 0.1863 ± 0.2563 |
> | G-learner     | 2.8681 ± 0.8696 | 0.2451 ± 0.1955 | 0.3200 ± 0.0086  | 0.0084 ± 0.0060 | 0.1691 ± 0.0622 | 0.0596 ± 0.0740 |
>
>
> **Weakness 5:**
> Explanation of Eq. (9).
>
> **Answer:**
> Eq. (5) is a Wasserstein barycenter between $\alpha$ and $\beta$ with the parameter $\kappa$.
> The barycentric projection considers the special case with $\kappa=1$, which obtains the representations of source samples {$x_{0,i}$} in the target distribution $\mu_1$.
>
> Following Remark 7.1 in [b] and Theorem 5.27 in [c], given an optimal coupling $\pi$, the barycenter is obtained by the push-forward interpolation $\mu_{\kappa} = P_{\kappa,\sharp} \pi$ where $P_t: (x_0, x_1) \in \mathbb{R}^d \times \mathbb{R}^d \mapsto (1-\kappa) x_0 + \kappa x_1$.
> For the discrete setup and the optimal transport matrix $\gamma^\*$, the interpolation is defined as
> $\mu\_{\kappa} = \sum\_{i=1}^{n\_0} \sum\_{j=1}^{n\_1} \gamma\_{ij}^{*} \delta((1-\kappa) x\_{0,i} + \kappa x\_{1,j})$.
> Please refer to Remark 7.1 in [b] and Theorem 5.27 in [c] for more details.
>
> **Weakness 6:**
> Discussion on Figure 3 in which the performance decreases as the number of intermediate groups increases.
>
> **Answer:**
> As the number of intermediate groups increases, much more samples are generated, which could amplify the influence of label noise, thereby resulting in a slight performance degradation
> Nevertheless, the results remain relatively stable across different values of $K$.
>
> [a] DESCN: Deep entire space cross networks for individual treatment effect estimation. SIGKDD 2022.
>
> [b] Computational optimal transport: With applications to data science. Foundations and Trends@ in Machine Learning, 11(5-6), 355-607.
>
> [c] Optimal transport for applied mathematicians, 2015.

---

### Official Review · Reviewer_z8oy · 2025-10-30

**Soundness:** 3
**Presentation:** 2
**Contribution:** 2
**Rating:** 2
**Confidence:** 4

**Summary:**

To balance the distributions between treated and control groups while reducing the loss of predictive information, the authors propose to generate intermediate groups through the Wasserstein geodesic, which smoothly connects the control and treated groups. The authors present a theoretical analysis of this method and demonstrate its effectiveness across several benchmark datasets.

**Strengths:**

1. The authors provide a theoretical analysis of the estimation error for their proposed method.

2. The authors focus an important issue in causal learning, treatment effect, and provide a potential solution to address confounding bias.

**Weaknesses:**

1. The language quality of the submission is not very good, especially in the abstract, which fails to clearly explain the motivation behind the proposed method.

2. Lacking ablation experiments to verify the effectiveness of each module.

3. The authors should compare one or some relatively new algorithms.

**Questions:**

1. The paper's writing style needs improvement, especially in the abstract and introduction, which fail to clearly explain the research motivation.

2. The novelty of this paper should be more clearly highlighted, enabling readers to easily understand the authors' contributions.

3. The authors should compare their work with some relatively new algorithms and conduct experiments on benchmark datasets, such as IHDP, for a more comprehensive evaluation.

4. This paper lacks a sensitivity analysis of the parameters to demonstrate the robustness of the proposed method.

5. The authors discuss the Optimal Transport in the related work section but fail to compare with it in the experiments.

6. Lacking ablation experiments to verify the effectiveness of each module.

---

> ### Author Response · Authors · 2025-11-27
>
> We sincerely appreciate the insightful feedback from the reviewers and present our detailed responses below. The manuscript will be revised accordingly based on both the reviewers’ comments and our replies.
>
> **Weakness 1 and Question 1:**
> Language quality and motivation.
>
> **Answer:**
> Thank you for your helpful comment.
> The major motivation is to address the overbalancing issue that arises in balanced representation learning.
> To tackle the imbalanced distributions of the control and treated groups caused by confounders, balanced representation learning usually adjusts confounders to align the distributions, which could remove predictive information about outcomes.
> To address this issue, we generate intermediate groups through the Wasserstein geodesic to smoothly connect the control and treated groups, thereby avoiding confounder adjustment and over-balancing.
>
> We will thoroughly revise the submission to highlight the motivation.
>
> **Weakness 2 and Question 6:**
> Ablation experiments to verify the effectiveness of each module.
>
> **Answer:**
> Thank you for your insightful suggestion.
> The ablation study results about G-learner without OT geodesic (w/o OT) or without data filtering (w/o filtering) are given as follows.
> The results demonstrate the effectiveness of the modules.
>
> | methods| News_in (pehe)  | News_in (ate)   | Twins_in (pehe) | Twins_in (ate)  | Jobs_in (pr)    | Jobs_in (att)   |
> | ------------- | --------------- | --------------- | --------------- | --------------- | --------------- | --------------- |
> | w/o OT        | 2.5735 ± 0.6322 | 0.2456 ± 0.0379 | 0.3199 ± 0.0022 | 0.0189 ± 0.0032 | 0.2663 ± 0.0296 | 0.0991 ± 0.0225 |
> | w/o filtering | 2.3738 ± 0.6638 | 0.2339 ± 0.1709 | 0.3197 ± 0.0020 | 0.0177 ± 0.0067 | 0.2218 ± 0.0554 | 0.1625 ± 0.2185 |
> | G-learner     | 2.2750 ± 0.7415 | 0.2258 ± 0.1472 | 0.3191 ± 0.0020 | 0.0107 ± 0.0039 | 0.2053 ± 0.0151 | 0.0552 ± 0.0306 |
>
>
> | methods         | News_out (pehe) | News_out (ate)  | Twins_out (pehe) | Twins_out (ate) | Jobs_out (pr)   | Jobs_out (att)  |
> | ------------- | --------------- | --------------- | ---------------- | --------------- | --------------- | --------------- |
> | w/o OT        | 2.8506 ± 0.8263 | 0.2832 ± 0.2541 | 0.3212 ± 0.0074  | 0.0243 ± 0.0107 | 0.2601 ± 0.0469 | 0.0902 ± 0.0783 |
> | w/o filtering | 2.8906 ± 0.8263 | 0.2948 ± 0.2417 | 0.3208 ± 0.0086  | 0.0140 ± 0.0067 | 0.1871 ± 0.0664 | 0.1863 ± 0.2563 |
> | G-learner     | 2.8681 ± 0.8696 | 0.2451 ± 0.1955 | 0.3200 ± 0.0086  | 0.0084 ± 0.0060 | 0.1691 ± 0.0622 | 0.0596 ± 0.0740 |

---

> > ### Author Response · Authors · 2025-11-27
> >
> > **Weakness 3 and Question 3**
> > Comparison with some relatively new algorithms and experiments on the IHDP dataset.
> >
> > **Answer:**
> > Thank you for your helpful suggestion.
> >
> > 1. We have added new algorithms CFR-Pro [a], CE-RCER [b], and DESCN [c] for comparison in the following.
> > Our proposed method achieves promising performance compared with others.
> >
> > | methods      | News_in (pehe)  | News_in (ate)   | Twins_in (pehe) | Twins_in (ate)  | Jobs_in (pr)    | Jobs_in (att)   |
> > | --------- | --------------- | --------------- | --------------- | --------------- | --------------- | --------------- |
> > | CFR-Pro   | 3.3859 ± 1.3643 | 0.5562 ± 0.4728 | 0.3197 ± 0.0023 | 0.0183 ± 0.0156 | 0.3007 ± 0.0216 | 0.0938 ± 0.0184 |
> > | CE-RCER   | 3.2342 ± 1.0884 | 1.3131 ± 0.4016 | 0.3204 ± 0.0027 | 0.0244 ± 0.0195 | 0.2886 ± 0.0378 | 0.1132 ± 0.0524 |
> > | DESCN     | 4.3748 ± 1.3147 | 2.6976 ± 0.7238 | 0.3174 ± 0.0031 | 0.0149 ± 0.0105 | 0.2811 ± 0.0349 | 0.1620 ± 0.0178 |
> > | G-learner | 2.2750 ± 0.7415 | 0.2258 ± 0.1472 | 0.3191 ± 0.0020 | 0.0107 ± 0.0039 | 0.2053 ± 0.0151 | 0.0552 ± 0.0306 |
> >
> > | methods     | News_out (pehe) | News_out (ate)  | Twins_out (pehe) | Twins_out (ate) | Jobs_out (pr)   | Jobs_out (att)  |
> > | --------- | --------------- | --------------- | ---------------- | --------------- | --------------- | --------------- |
> > | CFR-Pro   | 3.3837 ± 1.2978 | 0.5963 ± 0.4602 | 0.3206 ± 0.0088  | 0.0161 ± 0.0133 | 0.2935 ± 0.0623 | 0.0937 ± 0.0863 |
> > | CE-RCER   | 3.2956 ± 1.0875 | 1.2729 ± 0.4131 | 0.3214 ± 0.0095  | 0.0244 ± 0.0202 | 0.1781 ± 0.0401 | 0.1265 ± 0.1042 |
> > | DESCN     | 4.3308 ± 1.2460 | 2.6741 ± 0.6953 | 0.3244 ± 0.0081  | 0.0168 ± 0.0217 | 0.3011 ± 0.0509 | 0.1920 ± 0.0078 |
> > | G-learner | 2.8681 ± 0.8696 | 0.2451 ± 0.1955 | 0.3200 ± 0.0086  | 0.0084 ± 0.0060 | 0.1691 ± 0.0622 | 0.0596 ± 0.0740 |
> >
> >
> > 2. We have added experiments on the IHDP dataset as follows.
> > Our method maintains competitive performance on this dataset.
> >
> > |           | IHDP             |                  |                  |                  |
> > |-----------|------------------|------------------|------------------|------------------|
> > |           | pehe\_in           | pehe\_out          | ate\_in            | ate\_out           |
> > | EsCFR     | 1.1676 ± 1.1347  | 1.2581 ± 1.3166  | 0.2564 ± 0.3886  | 0.4048 ± 0.5693  |
> > | CFR\_wass  |  1.2879 ± 0.2508 | 1.3431 ± 0.2154  | 0.2158 ± 0.1583  | 0.3002 ± 0.1883  |
> > | CFR\_MMD   | 1.2075 ± 0.9271  | 1.3160 ± 1.0927  | 0.2104 ± 0.1267  | 0.3749 ± 0.2941  |
> > | DragonNet | 3.4911 ± 5.2079  | 3.2626 ± 4.8384  | 1.3069 ± 2.2908  | 1.9523 ± 3.5459  |
> > | GANITE    | 8.2262 ± 7.4578  | 5.5415 ± 1.2962  | 7.7728 ± 6.1049  | 6.4089 ± 3.7900  |
> > | DKLite    |  4.4072 ± 7.4140 |  3.7288 ± 5.7008 | 0.2953 ± 0.2497  | 0.9048 ± 1.8252  |
> > | G-learner | 1.1888 ± 0.7051  | 1.1952 ± 1.6739  |  0.3108 ± 0.2978 |  0.3738 ± 0.2959 |
> >
> > **Question 2:**
> > Highlight the novelty and contribution.
> >
> > **Answer:**
> > Thank you for your valuable comment.
> >
> > We will explicitly highlight the novelty and contribution in the revised version.
> > Most existing methods adjust covariates for distribution alignment, which could over-balance the covariates and lose predictive information.
> > Different from them, we instead adjust the outcome prediction model from one distribution to another, which avoids the issue of over-balancing.
> > To achieve this,
> > we bridge treated and control groups with intermediate distributions along the Wasserstein geodesic,
> > and gradually adapt the outcome model via self-training for strong generalization.
> > We further improve performance by high-quality sample filtering.
> > We theoretically analyze the property of our method, and empirically demonstrate the effectiveness of our method.
> >
> > Our novelty is also approved by reviewers.
> > Reviewer ttMD: "It has not been extensively studied in existing studies, and the authors' focus on this gap is highly relevant", "the authors' idea of incorporating optimal transport theory to address this challenge is both novel and well-motivated.", The authors' idea of incorporating optimal transport theory to address this challenge is both novel and well-motivated.
> > Reviewer c11B: "this paper proposes a novel method", "The use of intermediate domains to generate auxiliary samples for alleviating the over-balancing issue is both creative and effective.".
> > Reviewer JEDq: "Conceptual shift from 'adjut covariates' to 'adjust the prediction model' via a smooth geodesic path is a clear and appealing alternative to conventional balancing".
> >
> > **Question 4:**
> > Sensitivity analysis of the parameters.
> >
> > **Answer:**
> > Thank your for your insightful comments.
> > We provided the results of the sensitivity analysis in the appendix,
> > Section B.6 and Figure 3 on Page 16.
> > Our method can achieve promising performance with varying values of hyperparameters.

---

> > > ### Author Response · Authors · 2025-11-27
> > >
> > > **Question 5:**
> > > Comparison with optimal transport-based methods.
> > >
> > > **Answer:**
> > > Thank you for your helpful suggestion.
> > > Among the compared methods, ESCFR [d]  is a recent optimal transport-based methods.
> > > In addition, we have added optimal transport-based methods CFR-Pro[a], CE-RCFR[b], and DESCN[c] in **Answer** to **Weakness 3 and Question 3**.
> > >
> > > [a] Proximity Matters: Local Proximity Enhanced Balancing for Treatment Effect Estimation, SIGKDD 2025.
> > >
> > > [b] CE-RCFR: Robust Counterfactual Regression for Consensus-Enabled Treatment Effect Estimation, SIGKDD 2024.
> > >
> > > [c] DESCN: Deep entire space cross networks for individual treatment effect estimation, SIGKDD 2022.
> > >
> > > [d] Optimal transport for treatment effect estimation, NeurIPS 2023.

---

### Official Review · Reviewer_ttMD · 2025-10-30

**Soundness:** 3
**Presentation:** 4
**Contribution:** 4
**Rating:** 8
**Confidence:** 3

**Summary:**

In this paper, the authors address an important limitation of existing causal inference methods that rely on balanced representation learning: the over-balancing issue that arises in outcome prediction models due to imbalances in the treatment and control groups. They propose generating intermediate groups through the Wasserstein geodesic to adjust the outcome prediction model between consecutive groups via a self-training paradigm. This approach is novel and well-motivated, and supported by both theoretical and experimental evidence.

**Strengths:**

1. The problem addressed in this paper, the over-balancing issue that arises in outcome prediction models due to imbalances in the treatment and control groups, is indeed a significant and common challenge in the real-world application of causal inference methods based on balanced representation learning. It has not been extensively studied in existing studies, and the authors' focus on this gap is highly relevant.

2. The authors' idea of incorporating optimal transport theory to address this challenge is both novel and well-motivated. By leveraging the elegant properties of the Wasserstein geodesic, the approach enables the learning of intermediate distributions along the optimal transport paths between treatment and control groups. This, in turn, allows the outcome prediction model to be adjusted between consecutive groups via a self-training paradigm. Its ability to incorporate the control group distribution into the treatment group model (and vice versa) is an excellent solution to the imbalance problem.

3. The theoretical proof of the proposed method is quite thorough, providing a solid foundation for the approach.

4. The authors have conducted experiments on many benchmarks and included hyperparameter analysis, effectively demonstrating the effectiveness of the proposed method.

**Weaknesses:**

1. The current assumption that $p_0$ and $p_1$ follow uniform (or other prior) distributions seems challenging to satisfy in real-world scenarios, where such prior knowledge may not be available. I am curious whether it would be possible to directly estimate $p_0$ and $p_1$ from the data, thereby relaxing this assumption.

2. In the generated data filtering process, the authors select only the samples with high confidence. However, low-confidence samples might not only arise from model errors, but could also be inherently more difficult to learn due to their scarcity, such as in the case of long-tail distributions. I wonder whether the cumulative removal of these samples in each step could lead to the situation where such samples are never adequately learned by the model.

3. The paper aims to address the imbalances in the treatment and control groups, but the baselines compared do not specifically address this issue. I suggest comparing the proposed method with other baselines that also aim to resolve such imbalances, such as DESCN [1], to provide a more relevant evaluation.

[1] Zhong, K., Xiao, F., Ren, Y., Liang, Y., Yao, W., Yang, X., & Cen, L. (2022, August). Descn: Deep entire space cross networks for individual treatment effect estimation. In Proceedings of the 28th ACM SIGKDD conference on knowledge discovery and data mining (pp. 4612-4620).

**Questions:**

Please see Weaknesses. It is a very interesting work, and I look forward to the authors' responses and further discussions.

---

> ### Author Response · Authors · 2025-11-27
>
> We sincerely appreciate the insightful feedback from the reviewers and present our detailed responses below. The manuscript will be revised accordingly based on both the reviewers’ comments and our replies.
>
> **Weakness 1:**
> Marginal distribution estimation.
>
> **Answer:**
> Thank you for the insightful comments.
>
> 1. The uniform distribution is widely adopted in optimal transport [a][b].
> Without prior information of the underlying distribution, the uniform distribution can be derived from the principle of maximum entropy, which states that the probability distribution that best represents the current observations is the one with the largest entropy.
> The equal weights in the uniform distribution also reflect that each sample has the same importance.
>
> 2. We also try to obtain marginal distributions based on kernel density estimation, and report the results as follows.
> Overall, the two versions have similar performance. Due to the estimation difficulty of the marginal distribution, the version with uniform distributions performs slightly better.
>
> | methods       | News_in (pehe)  | News_in (ate)   | Twins_in (pehe) | Twins_in (ate)  | Jobs_in (pr)    | Jobs_in (att)   |
> | ----------- | --------------- | --------------- | --------------- | --------------- | --------------- | --------------- |
> | OT with KDE | 2.2904 ± 0.5661 | 0.3472 ± 0.2586 | 0.3205 ± 0.0024 | 0.0190 ± 0.0151 | 0.2145 ± 0.0610 | 0.1682 ± 0.0760 |
> | G-learner   | 2.2750 ± 0.7415 | 0.2258 ± 0.1472 | 0.3191 ± 0.0020 | 0.0107 ± 0.0039 | 0.2053 ± 0.0151 | 0.0552 ± 0.0306 |
>
> | methods       | News_out (pehe) | News_out (ate)  | Twins_out (pehe) | Twins_out (ate) | Jobs_out (pr)   | Jobs_out (att)  |
> | ----------- | --------------- | --------------- | ---------------- | --------------- | --------------- | --------------- |
> | OT with KDE | 2.8689 ± 0.7829 | 0.2434 ± 0.2959 | 0.3218 ± 0.0088  | 0.0222 ± 0.0170 | 0.1672 ± 0.0960 | 0.2277 ± 0.0514 |
> | G-learner   | 2.8681 ± 0.8696 | 0.2451 ± 0.1955 | 0.3200 ± 0.0086  | 0.0084 ± 0.0060 | 0.1691 ± 0.0622 | 0.0596 ± 0.0740 |
>
>
>
> **Weakness 2:**
> Discussion regarding data filtering.
>
> **Answer:**
> Thank you for the insightful question.
>
> 1. Indeed, generated samples with low confidence could be difficult cases.
> However, since our target is to adjust the outcome prediction model from one group to another, the reliability of the pseudo labels are vital for effective adjustment.
> Low-confidence samples could introduce label noises, which affects the performance of the prediction model.
> Similar approaches is also adopted in self-training [c].
> In order to better leverage low-confidence samples to tackle difficult cases such as long-tail distributions, one possible approach is to employ an active learning method to require human annotation [d].
>
> 2. We provide the ablation study of data filtering as follows.
> We observe that data filtering exhibits a substantial enhancement in performance.
>
> | methods         | News_in (pehe)  | News_in (ate)   | Twins_in (pehe) | Twins_in (ate)  | Jobs_in (pr)    | Jobs_in (att)   |
> | ------------- | --------------- | --------------- | --------------- | --------------- | --------------- | --------------- |
> | w/o filtering | 2.3738 ± 0.6638 | 0.2339 ± 0.1709 | 0.3197 ± 0.0020 | 0.0177 ± 0.0067 | 0.2218 ± 0.0554 | 0.1625 ± 0.2185 |
> | G-learner     | 2.2750 ± 0.7415 | 0.2258 ± 0.1472 | 0.3191 ± 0.0020 | 0.0107 ± 0.0039 | 0.2053 ± 0.0151 | 0.0552 ± 0.0306 |
>
> | methods         | News_out (pehe) | News_out (ate)  | Twins_out (pehe) | Twins_out (ate) | Jobs_out (pr)   | Jobs_out (att)  |
> | ------------- | --------------- | --------------- | ---------------- | --------------- | --------------- | --------------- |
> | w/o filtering | 2.8906 ± 0.8263 | 0.2948 ± 0.2417 | 0.3208 ± 0.0086  | 0.0140 ± 0.0067 | 0.1871 ± 0.0664 | 0.1863 ± 0.2563 |
> | G-learner     | 2.8681 ± 0.8696 | 0.2451 ± 0.1955 | 0.3200 ± 0.0086  | 0.0084 ± 0.0060 | 0.1691 ± 0.0622 | 0.0596 ± 0.0740 |
>
> 3. The sensitivity analysis of the hyperparameter $r$ presented in Appendix B.6 (page 16) further demonstrates that an appropriate data filtering ratio can substantially improve the performance.

---

> > ### Author Response · Authors · 2025-11-27
> >
> > **Weakness 3:**
> > Comparison with DESCN.
> >
> > **Answer:**
> > Thank you for your valuable advice.
> > In the following, we have added the results of the suggested baseline DESCN as well as other baselines (CFR-Pro [e] and CE-RCER [f]).
> > Our proposed method achieves promising performance compared with others.
> >
> > | methods      | News_in (pehe)  | News_in (ate)   | Twins_in (pehe) | Twins_in (ate)  | Jobs_in (pr)    | Jobs_in (att)   |
> > | --------- | --------------- | --------------- | --------------- | --------------- | --------------- | --------------- |
> > | CFR-Pro   | 3.3859 ± 1.3643 | 0.5562 ± 0.4728 | 0.3197 ± 0.0023 | 0.0183 ± 0.0156 | 0.3007 ± 0.0216 | 0.0938 ± 0.0184 |
> > | CE-RCER   | 3.2342 ± 1.0884 | 1.3131 ± 0.4016 | 0.3204 ± 0.0027 | 0.0244 ± 0.0195 | 0.2886 ± 0.0378 | 0.1132 ± 0.0524 |
> > | DESCN     | 4.3748 ± 1.3147 | 2.6976 ± 0.7238 | 0.3174 ± 0.0031 | 0.0149 ± 0.0105 | 0.2811 ± 0.0349 | 0.1620 ± 0.0178 |
> > | G-learner | 2.2750 ± 0.7415 | 0.2258 ± 0.1472 | 0.3191 ± 0.0020 | 0.0107 ± 0.0039 | 0.2053 ± 0.0151 | 0.0552 ± 0.0306 |
> >
> > | methods     | News_out (pehe) | News_out (ate)  | Twins_out (pehe) | Twins_out (ate) | Jobs_out (pr)   | Jobs_out (att)  |
> > | --------- | --------------- | --------------- | ---------------- | --------------- | --------------- | --------------- |
> > | CFR-Pro   | 3.3837 ± 1.2978 | 0.5963 ± 0.4602 | 0.3206 ± 0.0088  | 0.0161 ± 0.0133 | 0.2935 ± 0.0623 | 0.0937 ± 0.0863 |
> > | CE-RCER   | 3.2956 ± 1.0875 | 1.2729 ± 0.4131 | 0.3214 ± 0.0095  | 0.0244 ± 0.0202 | 0.1781 ± 0.0401 | 0.1265 ± 0.1042 |
> > | DESCN     | 4.3308 ± 1.2460 | 2.6741 ± 0.6953 | 0.3244 ± 0.0081  | 0.0168 ± 0.0217 | 0.3011 ± 0.0509 | 0.1920 ± 0.0078 |
> > | G-learner | 2.8681 ± 0.8696 | 0.2451 ± 0.1955 | 0.3200 ± 0.0086  | 0.0084 ± 0.0060 | 0.1691 ± 0.0622 | 0.0596 ± 0.0740 |
> >
> >
> > [a] Reducing Balancing Error for Causal Inference via Optimal Transport, ICML 2024.
> >
> > [b] Gromov-Wasserstein Averaging of Kernel and Distance Matrices, ICML 2016.
> >
> > [c] Understanding self-training for gradual domain adaptation, ICML 2020.
> >
> > [d] Causal-BALD: Deep Bayesian Active Learning of Outcomes to Infer Treatment-Effects from Observational Data, NeurIPS 2021.
> >
> > [e] Proximity Matters: Local Proximity Enhanced Balancing for Treatment Effect Estimation, SIGKDD 2025.
> >
> > [f] CE-RCFR: Robust Counterfactual Regression for Consensus-Enabled Treatment Effect Estimation, SIGKDD 2024.

---

### Meta-Review · Area_Chair_jGV1 · 2026-01-02

**Summary:**

The paper proposes a method called G-learner for causal inference. Unlike traditional methods that balance covariates (which can lead to over-balancing and loss of predictive information), this approach adjusts the outcome prediction model itself. The method uses Optimal Transport (Wasserstein geodesic) to generate intermediate distributions between the control and treated groups. It then employs a self-training paradigm to gradually adapt the prediction model across these intermediate domains, utilizing an uncertainty-based filtering mechanism to select high-quality samples.

The reviewers generally agreed that the problem of over-balancing is significant and the proposed solution (shifting from covariate adjustment to model adjustment via OT) is novel and interesting. Concerns primarily focused on the lack of specific baselines (particularly recent OT-based methods), the need for more diverse datasets (IHDP/ACIC), requests for ablation studies, and clarifications regarding the derivation and computational cost. The authors conducted extensive experiments to address those suggestions.

**Reviewer Concerns:**

Addressed Concerns:

- Reviewer z8oy (Score 2) and Reviewer c11B (Score 4) heavily penalized the paper for missing recent and relevant baselines. Specifically, c11B requested ESCFR-Pro and CE-RCFR , and z8oy requested comparisons to "relatively new algorithms". The authors fully addressed this. They added comparisons for CFR-Pro, CE-RCFR, and DESCN across News, Twins, and Jobs datasets. The results show G-learner performs competitively or better than these added baselines.


- Reviewer z8oy specifically asked for the IHDP benchmark. Reviewer JEDq asked for IHDP or ACIC. The authors successfully conducted and reported experiments on both IHDP and ACIC datasets in the rebuttal. The method showed competitive performance (e.g., lower PEHE on IHDP compared to GANITE and DragonNet).

- Reviewer z8oy requested ablations to verify module effectiveness. Reviewer c11B requested an ablation on data filtering. Reviewer JEDq asked about the necessity of OT geodesics vs. random interpolation. The authors provided comprehensive ablation tables.

- Reviewer c11B requested error bars for Figure 3, and z8oy requested sensitivity analysis. The authors provided the raw data for Figure 3 with standard deviations/error bars included. They also pointed z8oy to Appendix B.6 where sensitivity analysis was already present, and provided further clarification.

- The authors provided the Big-O complexity using the Sinkhorn algorithm and provided a runtime comparison table showing G-learner is faster than CFR_wass and DragonNet, though slower than GANITE.


Outstanding Concerns:

- The authors shall include all the results from additional experiments in their final version. The language (as Reviewer z8oy suggested) needs additional, significant polishing.

**Reviewer Scores:**

Reviewers ttMD and JEDq are likely to keep their scores.

Reviewer z8oy gave a 2 primarily due to missing ablations, missing baselines (IHDP), and missing sensitivity analysis. The authors provided every single item requested (new baselines, IHDP results, ablation tables). I expect Reviewer z8oy will increase his/her score to 4 or 6.

Reviewer c11B gave a 4 primarily due to missing specific OT baselines (CE-RCFR, etc.) and lack of error bars. The authors added the exact baselines requested and the error bars. I expect Reviewer c11B will increase his/her score to 6.

---

### Decision · Program_Chairs · 2026-01-26

Accept (Poster)